# Structure and unusual binding mechanism of the hyaluronan receptor LYVE-1 mediating leucocyte entry to lymphatics

Fouzia Bano [1,2,3,4,5,12,14], Suneale Banerji[6,14], Tao Ni [7,13,14], Dixy E. Green [8], Kalila R. Cook[1,2,3,4], Iain W. Manfield [4], Paul L. DeAngelis [8], Emanuele Paci[4,9], Martin Lepšík [10,11] ✉, Robert J. C. Gilbert [7] ✉, Ralf P. Richter [1,2,3,4,5] ✉ & David G. Jackson [6] ✉

Immune surveillance involves the continual migration of antigen-scavenging immune cells from the tissues to downstream lymph nodes via lymphatic vessels. To enable such passage, cells first dock with the lymphatic entry receptor LYVE-1 on the outer surface of endothelium, using their endogenous hyaluronan glycocalyx, anchored by a second hyaluronan receptor, CD44. Why the process should require two different hyaluronan receptors and by which specific mechanism the LYVE-1•hyaluronan interaction enables lymphatic entry is however unknown. Here we describe the crystal structures and binding mechanics of murine and human LYVE-1•hyaluronan complexes. These reveal a highly unusual, sliding mode of ligand interaction, quite unlike the conventional sticking mode of CD44, in which the receptor grabs free hyaluronan chain-ends and winds them in through conformational re-arrangements in a deep binding cleft, lubricated by a layer of structured waters. Our findings explain the mode of action of a dedicated lymphatic entry receptor and define a distinct, low tack adhesive interaction that enables migrating immune cells to slide through endothelial junctions with minimal resistance, while clinging onto their hyaluronan glycocalyx for essential downstream functions.

The migration of immune cells through lymphatic vessels is critical for immune surveillance and the subsequent generation and modulation of protective immune responses in draining lymph nodes[1–4]. In addition, the process facilitates the clearance of macrophages that remove macromolecular debris during the resolution of tissue injury[1,3,5]. A key receptor mediating both these processes is LYVE-1 (LYmphatic Vessel Endothelial receptor-1), an integral membrane adhesion molecule whose interaction with its glycosaminoglycan ligand hyaluronan (HA) in the surface glycocalyx of tissue migrating dendritic cells and macrophages enables their docking with the basolateral surface of initial lymphatic capillaries and transmigration to the lumen[6–9]. Located

within the specialised button-like endothelial junctions of such vessels[10–12], LYVE-1 makes the first adhesive contact with incoming immune cells through the formation of endothelial transmigratory cups[9,13]. The importance of the LYVE-1•HA axis for normal immune function has been underlined by the demonstration that its functional disruption by *lyve1* gene deletion, monoclonal antibody (mAb) blockade or HA depletion impairs the trafficking of antigen-loaded dendritic cells (DCs) to draining lymph nodes for priming of antigen-specific T cell responses[9]. In addition, it is now also evident that diurnal regulation of *lyve1* gene expression in peripheral lymphatics by the circadian clock system facilitates migration of DCs from tissues to lymph nodes

A full list of affiliations appears at the end of the paper. ✉e-mail: martin.lepsik@uochb.cas.cz; robert.gilbert@magd.ox.ac.uk; r.richter@leeds.ac.uk; david.jackson@imm.ox.ac.uk

during sleeping hours, when priming of T cell responses is most efficient[14]. Moreover, in response to tissue injury, LYVE-1 has been shown to mediate the clearance of inflammatory macrophages from the infarcted heart via epicardial lymphatics, a process critical for cardiac repair and limitation of subsequent fibrosis[15].

HA, the critical binding partner for LYVE-1, is synthesised by migrating DCs and macrophages where it forms a dense glycocalyx, tethered to the surface by CD44, a member of the hyaladherin family that is structurally related to LYVE-1[7,16,17]. Although both LYVE-1 and CD44 mutually engage HA and with broadly similar (micromolar) binding affinities, LYVE-1 exhibits a greater dependence on avidity, due to constraints in lateral mobility from corralling within the sub-membrane actin network and mutual charge repulsion imposed by sialylation[18–23]. Consequently, LYVE-1 requires surface clustering to enable multivalent binding to the long HA polysaccharide chains[7,8,16,24]. Secondly, unlike CD44, LYVE-1 also forms disulfide-linked homodimers which have a significantly higher HA-binding affinity than the monomer ($K_D$ ~8 μM compared to >100 μM)[6,8,25,26], due in part to a longer sugar-binding footprint (22 saccharide units in the homodimer compared to 8 in the monomer), evident from in vitro competition studies with HA oligosaccharides[21]. Thirdly, the character of the LYVE-1•HA adhesive interaction appears to differ from that of CD44[21,27], being exquisitely salt-sensitive and likely dominated by electrostatic interactions[8,21]. Lastly, CD44 is absent from lymphatic endothelium, although it can also anchor an HA glycocalyx on the luminal surface of blood capillaries, thereby facilitating adhesion and extravasation of lymphocytes and neutrophils that have high levels of the receptor but which, unlike DCs and macrophages, lack an HA coat[18,28,29]. Hence, the two receptors support HA-mediated cell trafficking, but in different ways and in different vasculatures. However, in the absence of a 3D structure for LYVE-1, it has been unclear until now how its distinct properties enable its dedicated function as a receptor for lymphatic entry.

Here, we have (i) investigated the physical nature of the LYVE-1 binding interaction with HA polymers by dynamic force spectroscopy, (ii) solved the crystal structures of the HA-binding domains (HABDs) in both mouse and human LYVE-1 and their HA ligand-bound complexes, and (iii) analysed the dynamics of the binding interaction via molecular dynamics (MD) simulations. Together, our studies reveal that LYVE-1 binds HA through an unusual sliding interaction distinct from that of CD44[30], in which the free ends of polymer chains are selectively engaged, clasped and progressively advanced through a flexible binding groove in the receptor, enabled by key conformational rearrangements within its surface and lubrication by a cushion of water-mediated hydrogen bonds. These properties endow LYVE-1 with the capacity for a selective and rapidly reversible mode of HA binding, whose unusual mechanics support the adherence, crawling and ingress of migrating immune cells from the outer surface of lymphatic capillaries to the lumen, in the low shear environment surrounding the lymphatic vasculature[16,31,32].

## Results

### LYVE-1•HA binding has unusual sliding characteristics

To explore the mechanics of HA binding by LYVE-1, we used dynamic force spectroscopy (DFS). Previous DFS analyses of CD44[30,33] revealed that the individual bonds holding HA within the receptor binding site are markedly resistant to rupture, despite their relatively low binding affinity ($K_D \approx 50$ μM[34]) and in keeping with the function of the receptor in stably anchoring an HA glycocalyx[3,9,35]. Here, we compared the HA binding and unbinding behaviours of human LYVE-1 and CD44 (hLYVE-1 and hCD44) when subjected to force using a planar interaction surface with C-terminally anchored receptor ectodomains and an atomic force microscopy (AFM) probe coated with reducing end-anchored high-$M_w$ HA chains (Fig. 1A and Supplementary Fig. 1). For CD44, detachment from HA was observed as a series of stochastic events

representing the sequential rupture of individual bonds in the force range of 30 to 60 pN (Fig. 1B, retraction curves shown in green). In marked contrast, the retraction curves for detachment of LYVE-1 from HA revealed collective bond rupture (Fig. 1C, in blue) and at lower forces (<30 pN), suggestive of multiple receptor molecules acting in concert to resist the pulling force exerted on the HA chain. As a possible explanation, we hypothesised that individual HA chains slide along their binding sites on adjacent LYVE-1 receptors under such force, detaching only when the chain termini are reached. In support of this interpretation, a dynamic model combining (i) multiple LYVE-1 receptors bound to an individual HA chain with the potential to stochastically translate in small steps along each binding site, and (ii) an entropically elastic model of the HA chain (Supplementary Fig. 2), faithfully reproduced the main features we observed in the experimental force curves (Fig. 1C, left inset). We termed this unusual and unexpected form of engagement a sliding interaction (i.e., motion along the ligand chain's linear contour) as distinct from the more conventional sticking interaction (i.e., engagement and release without motion along the chain) of HA with CD44 and all other HA binding proteins studied to date by DFS[30,33]. Of note, the recombinant LYVE-1 ectodomains used in these analyses were a mixture of monomers and homodimers, arising from interchain disulfide bonding at Cys201[26]. Nevertheless, similar mechanics were observed with the site-directed mutant hLYVE-1 Cys201Ala which forms exclusively monomers[26] (Supplementary Fig. 3), indicating the sliding interaction is an intrinsic property of the LYVE-1 HA-binding domain and independent of receptor homodimerisation.

Further DFS analyses confirmed that the unusual nanomechanical properties we observed for LYVE-1 are an inherent feature of the receptor. Firstly, both LYVE-1 and HA were strictly required for the interaction to be observed, and the interaction could be blocked with a function-blocking anti-LYVE-1 antibody. Secondly, varying either the surface density of immobilised LYVE-1 or the rate of tip retraction indicated the sliding mode of detachment from HA was maintained under each condition tested (Supplementary Fig. 4). Thirdly, the same dissociation pattern was preserved when LYVE-1 ectodomains were anchored on supported lipid bilayers (Supplementary Fig. 5) that more closely mimic the lateral mobility of the native receptor in the endothelial plasma membrane[36]. Lastly, the same general character of the force vs. distance curves was maintained with increasing salt concentration (from 150 mM NaCl up to 800 mM NaCl) despite a proportional reduction in the magnitude of binding (Supplementary Fig. 6). This reduction in binding, which we previously observed in conventional plate-binding assays[21] is indicative of a charge-dependent interaction between LYVE-1 and HA.

### LYVE-1 binds preferentially to non-reducing HA chain ends

The sliding behaviour observed during detachment of LYVE-1 from HA suggested the receptor initially binds to the chain ends rather than at internal sites along the polymer (Fig. 2). To test this directly, we investigated the binding behaviour of HA polymers containing consecutive 'closed' loops of varying lengths, generated by random internal biotinylation of the sugar chains before anchorage via streptavidin to the AFM tip (Fig. 2A). The results (Fig. 2B, C and Supplementary Figs. 7, 8) show that LYVE-1 can bind to such HA loops, requiring broadly similar forces (~40 pN) to those observed for detachment of CD44. Importantly however, the rates of such LYVE-1 'side-on' binding and unbinding were at least three orders of magnitude slower than those observed for free non-reducing ends, indicating LYVE-1 has a marked preference for binding to HA chain termini. Specifically, for side-on binding, we estimate $k_{on} = 5.5 \pm 3.0$ M$^{-1}$s$^{-1}$ from the frequency of DFS binding responses with HA loops (Supplementary Fig. 8), and $k_{off} \lesssim 0.002$ s$^{-1}$ from a fit to the data in Fig. 2C. These data imply a $K_D = k_{off}/k_{on}$ value on the order of a few 100 μM, consistent with previously published estimates for monovalent LYVE-1•HA

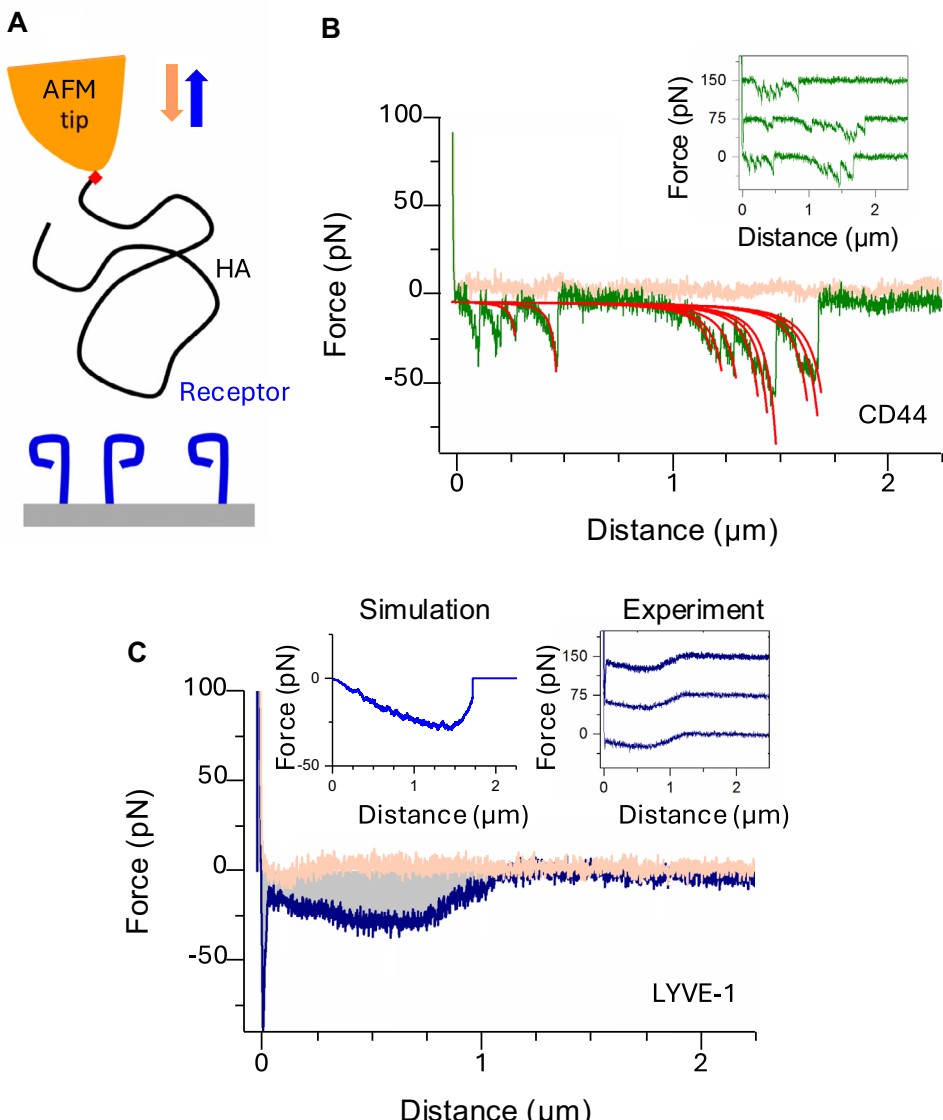

**Fig. 1 | The nanomechanics of LYVE-1•HA are distinct from CD44•HA and other biomolecular bonds. A** Schematic illustration of the dynamic force spectroscopy (DFS) setup to probe receptor•HA interactions (full wild-type ectodomains: hLYVE-1 Δ238 with a His$_{10}$ single tag and hCD44 Δ267 with a biotin/His$_{10}$ dual tag) with an HA chain (840 kDa) anchored via the reducing end (red diamond). Owing to the sharp AFM tip and the low HA grafting density, only one or at most a few HA tails can interact simultaneously with receptors on the substrate. **B** CD44•HA bonds are conventional sticking bonds. The representative force vs. distance curve (pink −tip approach, green−tip retract) shows a sequence of unbinding events representing the sequential and independent rupture of multiple CD44•HA bonds. Each unbinding event is well-fitted with a worm-like chain model with a 4.1 nm persistence length (red lines), confirming a single HA chain is being stretched. Three representative retract curves (inset, with $y$ axes offset for clarity) illustrate the stochastic nature of CD44•HA bond formation and sequential rupture along the HA chain. See Ref.[30] for a detailed analysis of CD44•HA bond mechanics. **C** LYVE-1•HA bonds are sliding bonds. The representative force vs. distance curve (pink−tip approach, blue−tip retract) reveals bond mechanics are unlike those of CD44•HA bonds. Three representative retract curves (right inset, with $y$-axis offset for clarity) illustrate the deterministic nature of LYVE-1•HA interactions, indicating that multiple receptors must act in concert on each HA chain. The gross shape and magnitude of the experimental retract curves are reproduced by a toy reductionist model (left inset; see Supplementary Fig. 2) that assumes HA stochastically moves in steps of one disaccharide and with a zero-force rate constant $k = 10^3 \, \text{s}^{-1}$ across the LYVE-1 binding sites and only detaches from a receptor once the chain end is reached. All data are representative of three independent experiments with distinct HA-coated probes and receptor-coated substrates per condition. Source data are provided as a Source Data file.

interactions[26] and with an independent analysis of the affinity by surface plasmon resonance for the specific LYVE-1 construct used here ($K_D = 227 \pm 76 \, \mu\text{M}$; Supplementary Fig. 9). By comparison, for end-on binding we conservatively estimate $k_{on} > 10^5 \, \text{M}^{-1} \, \text{s}^{-1}$, from the rapid HA binding (evidenced by an attractive force during approach; Fig. 1C and Methods) and hence $k_{off} > 10 \, \text{s}^{-1}$; and note that the exact rates may be up to two orders of magnitude higher than these values considering that sliding rates are on the order of $10^3 \, \text{s}^{-1}$ (Supplementary Fig. 2). Furthermore, the zero-force off rate for LYVE-1 binding to looped HA

($k_{off} < 0.002 \, \text{s}^{-1}$; Fig. 2C) was also much lower than that previously reported for CD44 ($0.6 \pm 0.1 \, \text{s}^{-1}$)[30], implying that the on-rate for LYVE-1 binding to looped HA is over two orders of magnitude lower than for CD44 and emphasising the marked contrast in the binding mode of these two receptors.

As the internally biotinylated HA chains used in these loop experiments still possess two free termini, we next tested whether both end-on and side-on binding can occur simultaneously on the same HA chain. Indeed, signatures of such behaviour could be

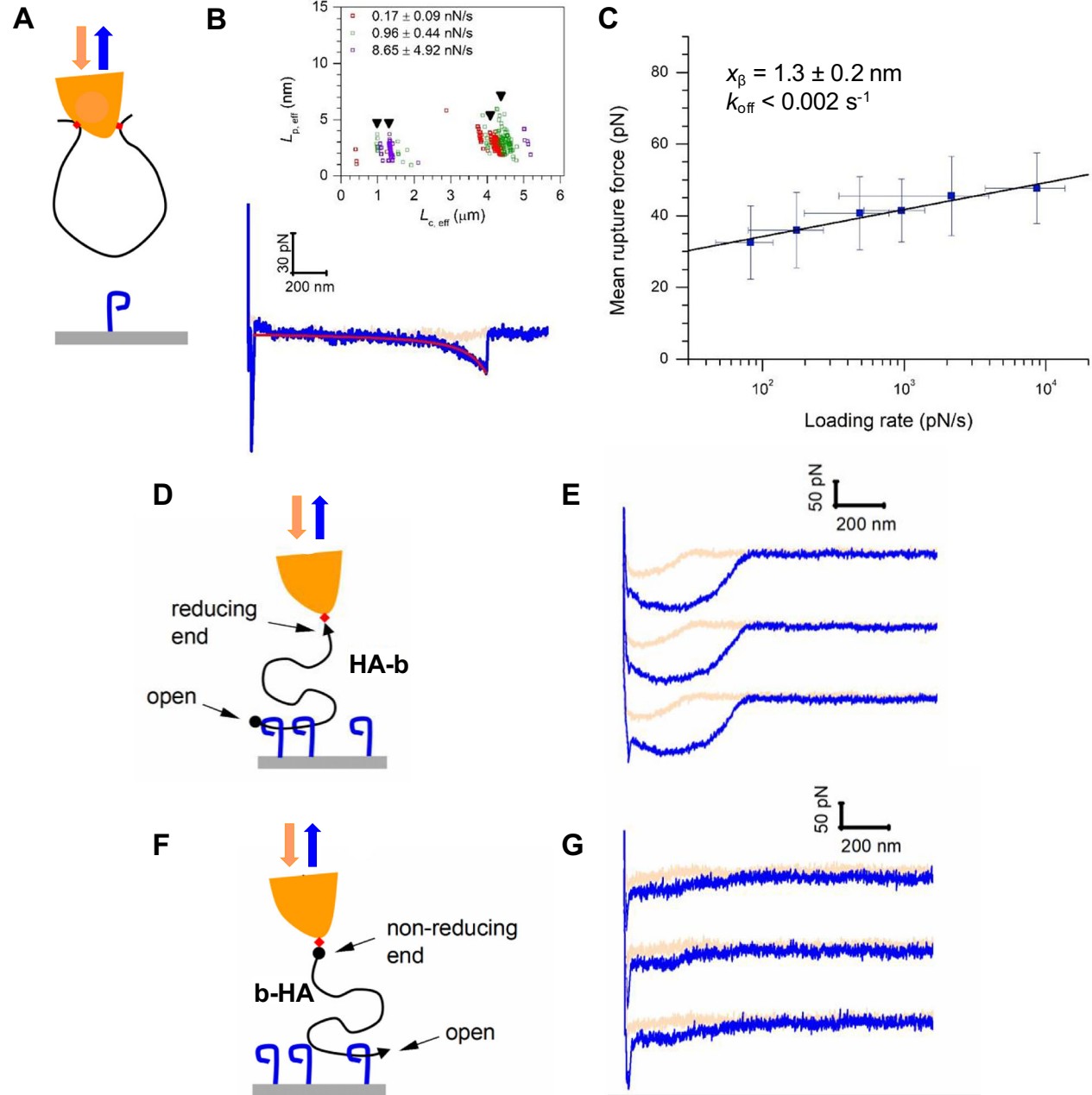

**Fig. 2 | LYVE-1 discriminates HA chains with a free end from closed loops and binds preferentially to the non-reducing terminus. A** Schematic illustration of the setup to probe interactions with closed HA loops where the low LYVE-1 receptor density (hLYVE-1 full wild-type ectodomain) enables probing of single bonds. Key results representative of three independent experiments with distinct HA-coated probes and LYVE-1 coated substrates are shown in (**B**, **C**), see Supplementary Figs. 7, 8 for a detailed analysis of the full dataset. **B** Representative force vs. distance curve (pink−tip approach, blue−tip retract) with a worm-like chain model fit (red line) to the unbinding event. The inset shows effective persistence lengths $L_{p,eff}$ vs. effective contour lengths $L_{c,eff}$ for three selected retract velocities (1, 4 and 12 μm/s, covering instantaneous loading rates as indicated with colour code as mean ± SD; a total of $n = 285$ data points are shown). The effective persistence lengths scatter just above 2 nm, consistent with the simultaneous and parallel stretching of two equal-sized HA chain segments. The effective contour lengths scatter around discrete values (vertical arrowheads indicate means), consistent with the stochastic probing

of a small set of loops of distinct size. **C** Mean rupture forces as a function of instantaneous loading rate (in semi-log presentation; mean ± SD; a total of $n = 482$ rupture events is included here, see Supplementary Fig. 7C for numbers resolved by loading rate) with a Bell-Evans model fit (black line; best fit (±1 σ confidence interval) parameters for the zero-force unbinding rate $k_{off}$ and the barrier width $x_\beta$ are displayed). **D**, **F** Setups to probe interactions with HA (320 kDa) immobilised via either the reducing (arrowhead; HA-b) or non-reducing (sphere; b-HA) ends, and **E**, **G** their representative force curves, offset along the y-axis for clarity (pink−tip approach, blue−tip retract). Both orientations show a response characteristic of deterministic binding to HA tails as in Fig. 1C. Note the magnitude of the force response is threefold weaker for HA with a free-reducing end (**G**), indicating pre-ferential binding to the non-reducing end. Forces displayed in (**G**) were increased by a factor of 1.9 to account for a proportionally reduced HA coverage of b-HA over HA-b (see Supplementary Fig. 11). Source data are provided as a Source Data file.

observed at higher LYVE-1 surface densities and when binding HA chains with a relatively low biotinylation ratio (Supplementary Fig. 10), indicating these present accessible termini that can engage sufficient numbers of LYVE-1 molecules to generate a measurable force response (cf. Supplementary Fig. 4B).

Finally, to establish whether LYVE-1 has a preference for binding an HA chain at the reducing or non-reducing terminus (i.e., reflecting the sugar polymers chain directionality), we performed further DFS and QCM-D analyses using identically sized (~320 kDa) polymers, immobilised by a single biotin handle located exclusively at one or the other end (Fig. 2D, E and Supplementary Fig. 11). Intriguingly, the resulting DFS retraction curves showed that the magnitude of the force response was some threefold greater for chains with a free non-reducing end, and the dissipation shifts showed a much reduced initial binding to HA via the free-reducing end. Together, these findings point to a distinct HA binding and unbinding mode for LYVE-1, in which the non-reducing end of the sugar polymer preferentially docks and subsequently threads through serial adjacent receptors via a sliding interaction that supports rapid and reversible adhesion.

## LYVE-1 HABD crystal structures show a deep binding groove
We expressed soluble extracellular domain constructs of the murine and human LYVE-1 HABDs as C-terminal histidine-tagged proteins in CHO cells for X-ray crystallography. The resulting proteins (mLYVE-1 Δ143 and hLYVE-1 Δ144) comprised the conserved HA-binding Link module, its flanking β0 and β7 strand extensions and the three conserved intramolecular disulfide bridges predicted from primary sequence comparisons with mCD44 and hCD44[27,34,37]. As highlighted by the sequence alignments in Supplementary Fig. 1, there is a high degree of homology between mLYVE-1 and hLYVE-1 as well as significant differences in the identities of some ligand binding residues. To allow us to investigate these further and obtain a comprehensive picture of the LYVE-1•HA binding interaction, we carried both species forward for structural analyses.

Crystals diffracted to 1.54 Å (mLYVE-1) and 1.64 Å (hLYVE-1). The hLYVE-1 apo structure was solved using sulfur single-wavelength anomalous dispersion (SAD) (see Methods), and then new datasets for both the hLYVE-1 and mLYVE-1 structures were solved by molecular replacement using the initial hLYVE-1 model (Supplementary Table 1, Supplementary Fig. 12 and Fig. 3A, B). The two LYVE-1 forms are very similar (RMSD = 0.68 Å for 110 Cα pairs, 1.04 Å across all 115 pairs) each containing a disulfide-bonded Link domain of six β strands and two α helices, flanked by N- and C-terminal extensions and a third disulfide bridge, much as in CD44 (Fig. 3C and sequence alignment in Supplementary Fig. 1). However, the HABD in LYVE-1 is more compact, due both to the first β-strand (β0) of the Link extension being shorter, and most obviously, the last two strands (β8 and β9) being absent; their corresponding sequences code for a serine/threonine-rich tract that is predicted to be unstructured[8,21,37] and was thus omitted in construct design (Fig. 3 and sequence alignment in Supplementary Fig. 1). Among other features, the β0 strand in human LYVE-1 is preceded by a turn of α-helix not resolved in the mouse structure, while the two N-linked glycan sidechains associated with in vivo regulation of HA-binding (on mAsn52/159 and hAsn53/160) were resolved in both forms.

Most striking however, is the deep binding groove in the LYVE-1 HABD structures (Figs. 3, 4), with the over-arching sidechains of residues mLys107/hLys108 and mArg104/hLys105 (Fig. 4) projecting from the β4/β5 loop, itself braced in position by the disulfide bond mCys84-Cys105/hCys85-Cys106. Additionally, mTyr86/hTyr87 and mTrp115/hTrp116 form the lower edge of the groove, beneath mLys107/hLys108 (Fig. 4), confirming the involvement of these residues in HA contact, as previously predicted from site-directed mutagenesis[21]. Although an analogous β4/β5 loop is also prominent in the CD44 HABD, it is less bulky and overlies a far shallower groove in what is a much wider and more open binding surface (Fig. 3C). Another key difference between

the LYVE-1 and CD44 HABDs is in the electrostatic properties of their surfaces (Fig. 3). The HA-binding clefts of human LYVE-1, and to a greater extent mouse LYVE-1, present a highly concentrated distribution of positive charge (Fig. 3), while the relevant surface of CD44 is rather more neutral and its most basic patch is far remote from the HA binding position[27].

## The HA chain is caught deep within the LYVE-1 binding groove
We also co-crystallised the mLYVE-1 construct Δ141 (Supplementary Fig. 1) with a synthetic HA octasaccharide (HA8) and the hLYVE-1 construct Δ144 with a decasaccharide (HA10). The structures which were solved at resolutions of 1.05 Å (mLYVE-1) and 1.32 Å (hLYVE-1), each yielded well-defined electron density for six of the bound sugar rings (see density maps in Supplementary Fig. 12), constituting GlcNAc2–GlcUA7 in the mouse complex, and most likely GlcNAc4-GlcUA9 in the human complex, numbered from the non-reducing end (i.e., the left hand side of the chain as viewed in Fig. 4A, B). The apparent advancement of the longer sugar chain by two residues in the hLYVE-1 HABD is not only consistent with the sliding behaviour observed in DFS but also leaves a single unbound GlcNAc overhang at the reducing end (position 10) in common with the mLYVE-1 complex. This is unlike the mCD44 HABD, where there is no such overhang, as the final GlcNAc is instead held within the binding groove[27].

As is clear from the structures of both LYVE-1 complexes (Figs. 5, 6) the bound HA hexasaccharide unit is clasped between the over-arching β4/β5 loop and the opposing surface of the positively charged binding cleft (Fig. 3 and Supplementary Fig. 13). With key contributions from the prominent mArg104/hLys105 and mLys107/hLys108 residues of the loop at its ceiling, and mTyr86/hTyr87 and mTrp115/hTrp116 at its floor, the HA binding surface is formed predominantly from mIle96/hIle97 and the mCys84-Cys105/hCys85-Cys106 disulfide (Figs. 5, 6), yielding a wide contact area of 462 Å² in the case of mLYVE-1 and 465 Å² for hLYVE-1, that is considerably larger than that of CD44 (372 Å² for crystal form A, 2JCQ and 400 Å² for crystal form B, 2JCR, see Methods for calculation). The bound HA is also held by hydrophobic residues that make respectively six contacts with sugar in the mouse complex (between Asn102/Cys105, Cys84/Cys105 and Ile96/Trp115) and seven in the human (between Asn103/Cys106, Ile97/Cys85/Cys106 and Ile97/ Trp116; Supplementary Table 2). Combined with the depth of the binding groove, this results in a buried surface area for HA that is some 14% larger than in CD44. Of note, it is apparent that HA binding results in more rigid LYVE-1 structures, as reflected in the higher structural resolutions obtained for the LYVE-1•HA complexes, and their reduced crystallographic temperature factors, compared to CD44 (Supplementary Fig. 14).

## Molecular dynamics reveal flexibility within the LYVE-1 HABD
We next assessed the flexibility of the LYVE-1 HABD pre- and post HA binding using molecular dynamics (MD) simulations of the mouse and human apoproteins and their HA-bound complexes. While these revealed significant fluctuations in the N- and C-terminal regions (i.e., β0, β1, and β6 strands), by far the greatest movements were observed within the clasp-like β4/β5 loop, most notably in the mouse receptor (see calculated root mean square fluctuations (RMSFs, Supplementary Fig. 15). As might be expected, the flexibility of the key contact residues mArg104/hLys105 and mLys107/hLys108, along with mTyr86/hTyr87 and mTrp115/hTrp116 decreased upon HA binding. Nevertheless, the extent of the reduction (expressed as RMSF$_{apo}$/RMSF$_{complex}$ ratio) was relatively small (average of 1.2 and 1.7 for mLYVE-1 and hLYVE-1, respectively) and in the case of mLys108/hLys107, the flexibility remained almost the same or was even increased (Supplementary Table 3A). By contrast in CD44, HA binding imposed a much greater reduction in flexibility, with RMSF$_{apo}$/RMSF$_{complex}$ average ratios of 2.5 for the analogous HA contact residues Arg45, Tyr46, Arg82, Tyr83, and Tyr109 (Supplementary Table 3B). Furthermore, the bound HA chain

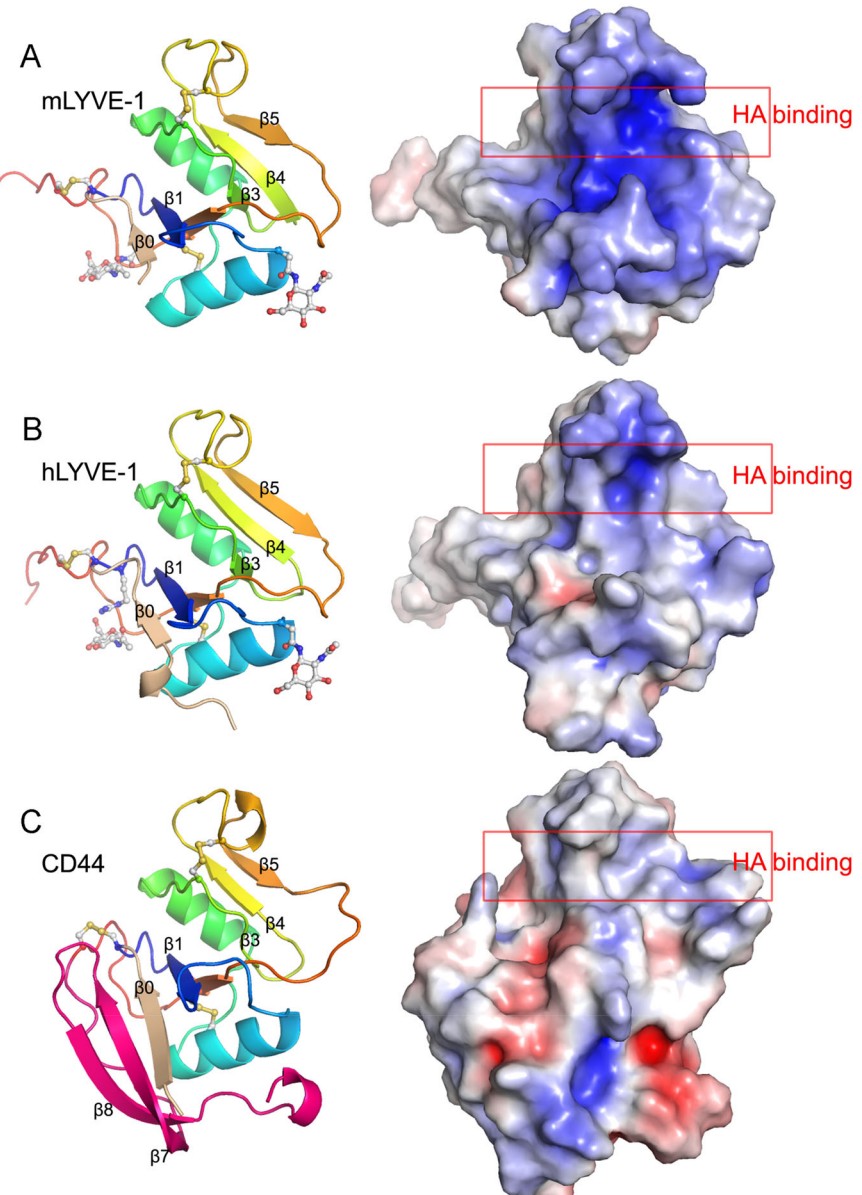

**Fig. 3 | X-ray crystallographic structures of unbound mLYVE-1 and hLYVE-1 HABDs compared to the mCD44 HABD. A** mLYVE-1, **B** hLYVE-1, **C** mCD44. On the left in each case, shown as a ribbon representation with the three disulfides present in each HABD shown in ball-and-stick format, as well as the glycan sidechains at mAsn129/hAsn130 (lefthand in this view) and mAsn52/hAsn53 (righthand in this view). Glycans were not present in the CD44 crystal structure as the protein was expressed and purified from *E. coli*. Prominent β strands within the structures are labelled, and coloured rainbow wise from the amino terminal to the carboxy terminal, save for β0 (*wheat* coloured in each case) and β7 and β8 in CD44 (pink). On the right in each case is a surface electrostatics representation scaled −5.0 $k_BT/e$ (red) to +5.0 $k_BT/e$ (blue) computed using APBS[92] and with the HA binding site identified previously[21,27] boxed. Backbone variations between CD44 and LYVE-1 were mLYVE-1 RMSD 0.92 Å for 90 Cα pairs, and 5.2 Å across all 113 pairs; hLYVE-1 RMSD 0.87 Å for 89 Cα pairs, and 4.04 Å across all 115 pairs. The greater variance in the all-pairs comparisons is due principally to the long loop between β5 and β6.

retained more flexibility in mLYVE-1/hLYVE-1 than in CD44 (average RMSF of 1.7 Å/2.0 Å vs. 1.5 Å, respectively; Supplementary Table 3C).

**Conformational changes in the HABD enable end-on HA binding**
Comparison between the LYVE-1 apoprotein and HA-bound complexes indicates that docking of the sugar occurs most likely through induced fit, achieved by conformational changes in key amino acid sidechains within the binding groove (Supplementary Movies 1, 2). Specifically, these relieve a block in the binding groove at its upper and lower edges to allow sequential docking of the HA non-reducing terminus and its likely directional advance along an unimpeded binding surface (Figs. 5, 6). The alternative conclusion, that HA binding involves conformational selection, although possible, seems less likely given the

particularly high resolution of the unbound LYVE-1 structures and their implied conformational stability.

In mLYVE-1, HA-induced unblocking of the binding groove is achieved by an upward flip of mArg104 in the clasp-like β4/β5 loop, away from its position obstructing the distal end and a sideways movement of mLys107 towards mTyr86 and mTrp115 at the groove lower edge (Fig. 5 and Supplementary Movie 1). By contrast, in hLYVE-1, unblocking involves a smaller lateral movement by hLys108 in the groove's upper edge, but more critically, the lowering of the hTrp116 and hTyr87 sidechains on the groove's opposite side (Fig. 6 and Supplementary Movie 2). The critical importance of these residues for HA binding is further underscored by the consequences of their site-directed mutagenesis, which previously identified hTyr87 and hTrp116

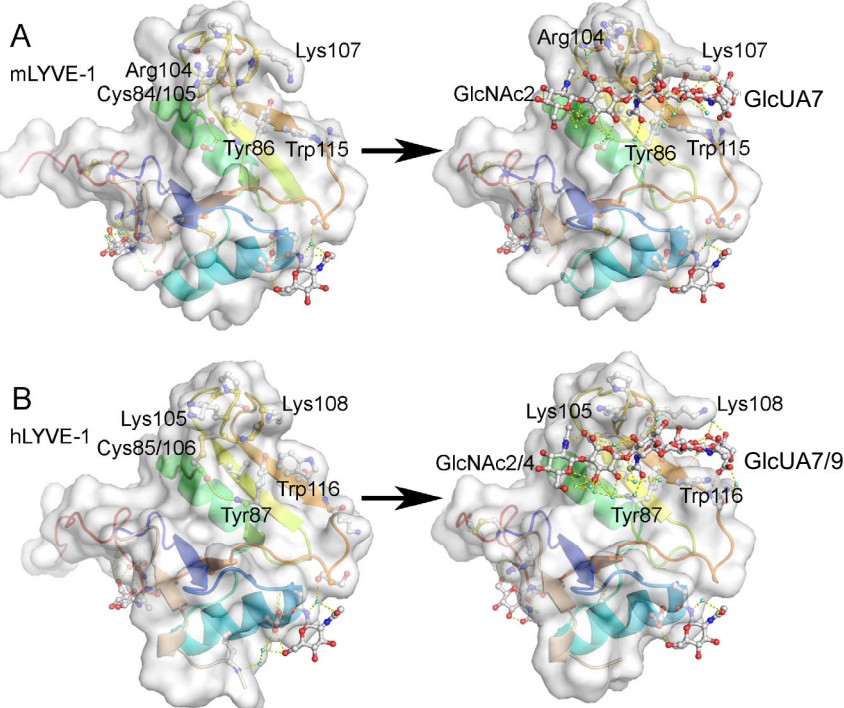

**Fig. 4 | Structures of the free and HA-bound complexes of mouse and human LYVE-1 compared.** Surface representations of the unbound (left) and HA-bound (right) structures of mLYVE-1 (**A**) and hLYVE-1 (**B**), with key residues in the HA binding site labelled. Significant conformational changes are limited to the HA binding groove. See the main text for more details.

as components of the epitope for the hLYVE-1 HA-blocking mAb 3A [38] (Supplementary Fig. 16).

## Dynamic H bonds and a water cushion hold HA and support sliding

The crystallographic structures of the LYVE-1•HA complexes show that hydrogen bonding contributes extensively to HA binding. Specifically, these indicate a mixture of direct H bonds (7 in mLYVE-1 and 5 in hLYVE-1) and indirect water-mediated H bonds (7 in mLYVE-1 and 8 in hLYVE-1; Figs. 5, 6 and Supplementary Table 2), the small differences in their numbers contributing to the slight difference between the register of HA in the binding groove of the mouse and human receptors (Figs. 5C, 6C and Supplementary Table 2). Four of the water molecules involved in indirect hydrogen bonding (w4-w6 and w10) are conserved in both the mouse and human LYVE-1•HA complexes (Figs. 5B, 6B and Supplementary Table 2), and none of these is observed in the apo-protein structures, apart from isolated examples associated with the *N*-linked glycan sidechains (mAsn52/129 and hAsn53/130) that lie far from the binding groove. Two further water molecules, w2 and w7 are conserved in position between the murine and human structures but are not engaged in the same interactions in each of the two complexes. Importantly, none of the waters is involved in crystal contacts, precluding the possibility they represent artefacts of crystallisation. These features highlight how binding of the HA chain to mouse and human LYVE-1 fixes waters in situ, similar to the fixation of water molecules by their *N*-glycan sidechains (Figs. 5B, 6B, Supplementary Fig. 17, and Supplementary Table 2). Such abundance of water-mediated hydrogen bonding is not found in CD44, where the HA chain is instead held by only two such bonds, with 11 direct H bonds as well as a more limited set of hydrophobic interactions than in LYVE-1.

Further analysis of the hydrogen bonding network by MD revealed that the direct H-bonding between the LYVE-1 receptors and HA is rather dynamic, with occupancies ranging from 24 to 82% and in numerous cases found only in two of three independent MD trajectories (Supplementary Table 4A). Exceptions to this observation were two very stable H bonds in the case of mLYVE-1 with occupancies exceeding 90%. This contrasts with CD44, in which nearly half of the direct H bonds were very stable (Supplementary Table 4A). Regarding the water-mediated hydrogen bonds, there were no cases of very stable interactions and the occupancies ranged from 11 to 83% (Supplementary Table 4B). Intriguingly, the dynamical nature of the calculations enabled us to visualise chains of water molecules lubricating the LYVE-1•HA interface (Supplementary Fig. 17A, B). This unusual feature contrasts with CD44, in which only two water molecules bridge HA to the binding groove (Supplementary Table 2), with calculated occupancies in MD of 45 and 11% (Supplementary Table 4B) and with no observed water molecule chains (Supplementary Fig. 17C).

We note that in a few cases of direct or water-mediated interactions, the crystallographic H bonds were not reproduced by MD (Supplementary Table 4). This is most likely the consequence of an erroneous overestimation of charged interactions inherent in the use of non-polarisable force fields in these challenging, highly charged systems[39,40]—a recognised shortcoming of the methodology, which in turn slightly distorted the crystallographic binding modes.

The presence of a bound water layer appears to be exclusive to LYVE-1 amongst known HA binding proteins. We hypothesise that the indirect interactions mediated by this water cushion in LYVE-1 provide lubrication for the observed sliding mode of HA engagement, by enabling more rapid binding and unbinding from the receptor.

## A coherent molecular basis for the LYVE-1 sliding interaction

The threading of the HA polymer tail through the groove beneath the β4/β5 arch in LYVE-1 provides a plausible mechanism for its preferential binding to the free non-reducing end of the HA chain as implicated in the DFS studies, whereby the chain end displaces sidechains of mArg104, mLys107/hLys108, hTrp116 and hTyr87 as it enters

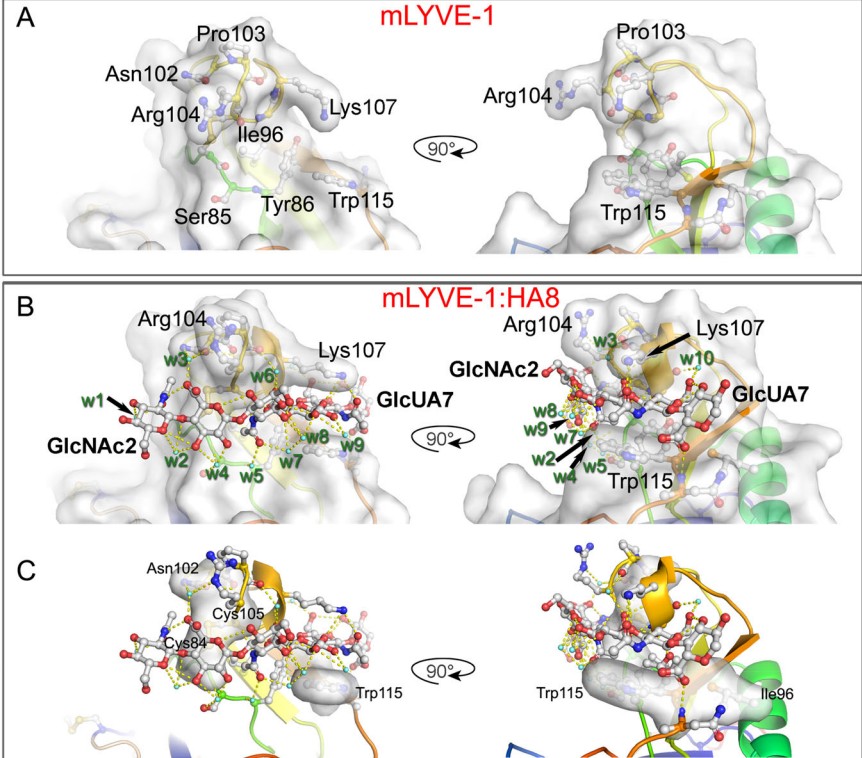

**Fig. 5 | Key residues and water-mediated interactions holding HA within the mLYVE-1 binding surface.** Close-ups of the HA binding groove in the mouse receptor with key gatekeeper and ligand residues labelled. In each case, orthogonal views are shown as indicated by the arrow. **A** mLYVE-1 apoprotein structure, **B**, **C** mLYVE-1•HA8 complex. In **B**, **C** water molecules resolved and involved in either bridging interaction between HA and LYVE-1 or bridging two or more atomic sites on HA are shown as aquamarine spheres, and hydrogen bonds are indicated by dashed yellow lines. Water molecules are labelled w1-w17 (of which w1–w10 are present for mLYVE-1) – see text for more details and Supplementary Table 2. Atomic surfaces are all protein atoms in (**B**) and hydrophobic contacts are rendered as a surface in (**C**). See also Supplementary Movie 1.

the binding groove (see Supplementary Movies 1, 2). It is important to note that the alternative side-on binding of HA would sterically clash with these pivotal sidechain movements, even if conformational selection rather than an induced fit were involved−a consequence that may well explain its much slower rate compared to end-on binding (Fig. 2C). This contrasts with CD44, where the absence of such steric hindrance enables side-on binding of HA, stabilised by the upswing of a pivotal arginine sidechain (Arg 41/45 in mCD44/hCD44, see Supplementary Movie 3). Also apparent from both the mouse and human LYVE-1•ligand complexes is that having bound the non-reducing end of the HA chain, the receptor slides along by two sugar rings at a time to maintain the same register – an action that helps explain how multiple receptors can thread along the same HA chain to harness the known avidity-dependent nature of the LYVE-1 binding interaction. Furthermore, the large proportion of the HA binding interface (25%) contributed by hydrophobic interactions, combined with the rapid exchange between structured waters within the H-bonding network provide a basis for both binding of the HA chain in register as well as lubrication for its sliding. Within the physiological context (Fig. 7), we envisage this sliding action enables migrating immune cells bearing a HA glycocalyx to crawl along the basolateral surface of LYVE-1 lined lymphatic capillaries and transmigrate to the lumen, in the low shear conditions of the surrounding tissue matrix.

## Discussion

Here we have described the structure and unusual properties of LYVE-1, the receptor in lymphatic capillaries that binds the endogenous HA glycocalyx of dendritic cells and macrophages and mediates their transit from the tissue interstices to the vessel lumen for subsequent migration to downstream lymph nodes[9,16]. Although closely related to

CD44, the receptor that anchors the glycocalyx in such cells[16], it has been unclear until now how LYVE-1 mediates lymphatic entry, how its binding properties differ from CD44 and how the mutual interaction of these two distinct receptors with HA supports such a role. Here, we have addressed these questions through our analyses of LYVE-1•HA binding mechanics and structures of the LYVE-1 HA-binding domain and its ligand-bound complexes, allowing us to define the molecular basis for LYVE-1 function.

Regarding mechanics, our DFS analyses revealed that LYVE-1 binds far more rapidly to the free ends of HA chains than internally (i.e., side-on binding), with a preference for non-reducing HA termini, and that the bound HA chain can advance through consecutive LYVE-1 molecules by means of a sliding motion, and subsequently retract through collective unbinding. These properties are very different to CD44, which displays side-on binding to HA and sequential detachment through individual bond breakage. Indeed, to our knowledge, they are unique to LYVE-1 among known HA receptors. Curiously, however, in the case of mammalian and streptococcal hyaluronidases, an analogous sliding interaction with HA occurs, which means the sugar chain is degraded processively through advancement in the non-reducing to reducing-end direction after random internal cleavage of a β1-4 GlcUA-GlcNAc linkage[41–43]. In addition, directional binding and sliding have been suggested for sequestration of the growth factors FGF-1 and FGF-2 by chains of the related glycosaminoglycan heparan sulfate[44–46], and for the advancement of thrombin along chains of heparin during the formation of its tripartite anticoagulant complexes with antithrombin respectively[47,48]. These sliding ligand interactions are analogous to processive enzymes with linear substrates such as cellulases[49] and some DNA and RNA polymerases[50,51].

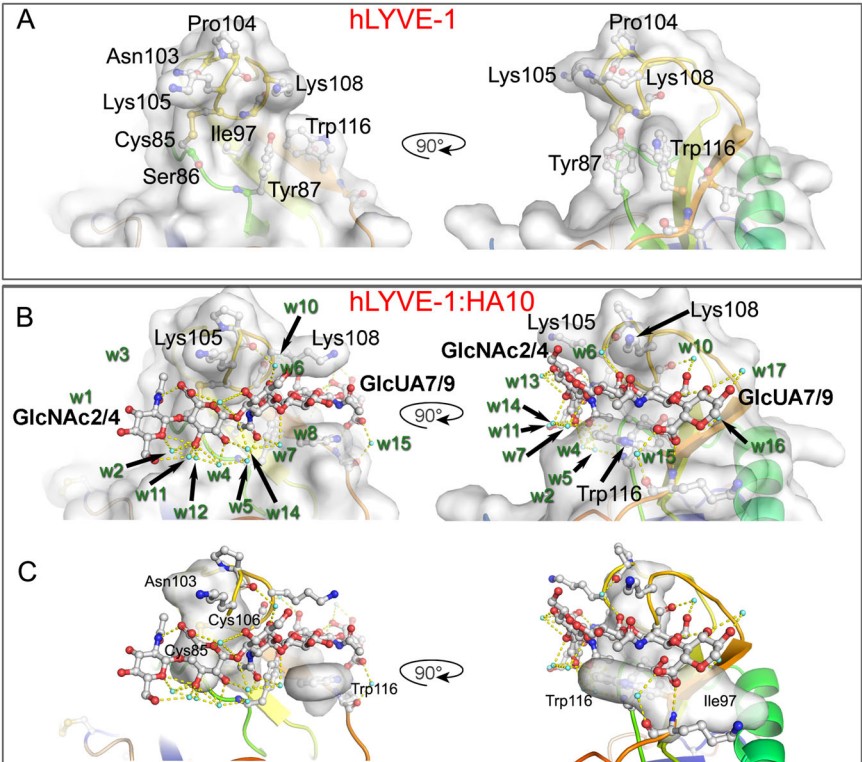

**Fig. 6 | Key residues and water-mediated interactions holding HA within the hLYVE-1 binding surface.** Close-ups of the HA binding groove in the human receptor with key gatekeeper and ligand residues labelled. In each case, orthogonal views are shown as indicated by the arrow. **A** hLYVE-1 apoprotein structure, **B** hLYVE-1•HA10 complex. Resolved water molecules involved in HA: hLYVE-1 or HA:HA interactions are shown as aquamarine spheres with numbering as in Fig. 5, and hydrogen bonds are similarly indicated by dotted yellow lines – see text and Supplementary Table 2 for more details. In **C**, the ten bound waters considered an index in mLYVE-1 are extended with seven additional waters observed only in the human receptor. Atomic surfaces are all protein atoms in (**B**), and hydrophobic contacts are rendered as a surface in (**C**). See also Supplementary Movie 2.

Our analyses of the murine and human LYVE-1 crystal structures identified the binding surface as a closed pincer-like groove into which the sugar chain docks via its non-reducing end and advances through co-ordinate displacement of arginine and/or lysine sidechains in the groove's upper edge, and rotational movement of tryptophan and tyrosine sidechains in its floor (Supplementary Movies 1, 2). The resulting ratchet-like binding arrangement in LYVE-1 differs considerably from the relatively open and unobstructed HA binding groove in CD44, where the upswing of a single key arginine residue (mArg45/hArg41) toward its lower face occurs after docking of the sugar and serves instead to consolidate the interaction[27,34] (Supplementary Movie 3). In addition, and consistent with their distinct binding modes, the types of interaction co-ordinating HA in the binding groove of LYVE-1 differ significantly from those in CD44. Along with a greater number of hydrophobic interactions and fewer direct H bonds in LYVE-1, the most prominent feature was an extensive network of structured waters located above and below the sugar protein interface, forming numerous indirect H bonds with the HA chain in dynamic exchange as indicated in our MD simulations. The contribution of such a large network of structured waters to the HA binding interface is unprecedented amongst HA receptors and hyaluronidases, and we know of no similar examples[41]. In other proteins such as the *H. influenzae* sialic acid binding protein SiaP and the G-protein coupled opioid receptors and rhodopsins, such water networks are thought to contribute to ligand binding[52], conformational dynamics and signal transduction, respectively[53,54], whereas in the sliding clamp proteins of DNA polymerase complexes, similar binding to structured waters has been described as 'water skating'[55]. In LYVE-1, the capacity of the HA chain to first displace and then form dynamic H bonds with the surface water layer is consistent with its function as a lubricating cushion around the sugar. Allied to this, both the main contact residues and the bound HA chain displayed a high degree of flexibility in the HA binding cleft, considerably more so than in CD44. Collectively, this combination of protein structural dynamics and water-mediated H-bonding in LYVE-1 is likely responsible for the distinctive sliding interaction of HA as it engages with the sugar-binding groove.

It is neither fully clear how the non-reducing ends of HA chains are organised, nor how easily they are accessible for capture by LYVE-1, when embedded within the surface coat of an immune cell, already complexed with their anchoring receptor CD44. As we reported recently, the CD44-bound HA glycocalyx of murine migratory DCs forms a thick coating around the cell that extends some ~500 nm from the plasma membrane as visualised by staining with the bVG1 probe and confocal (Airyscan) microscopy[16]. Notably, however, when DCs adhere to lymphatic endothelium in vitro this circumferential distribution of CD44 is altered to form a dense polar cap at the uropod– the dynamic region at the rear of the DC involved in cell:cell adhesion[16,24]. We anticipate the increase in CD44:HA binding stoichiometry that would be generated by such redistribution of the receptor would lead to an increase in local density of free non-reducing HA chain ends, and that selective binding of these termini by LYVE-1, aided by slower side-on binding to HA loops initiates docking of the DCs to endothelium. Subsequent transmigration would then proceed with the HA glycocalyx being retained by the DC, but peeled from the endothelium through collective unbinding and reverse sliding of the sugar chains on LYVE-1, aided by stochastic bond rupture/re-formation on CD44, under the tractive forces exerted by chemokine directed and actomyosin driven DC locomotion (see Fig. 7). Maintaining attachment of the HA glycocalyx by DCs in this way is likely to be important for their intraluminal crawling behaviour, as observed by intravital

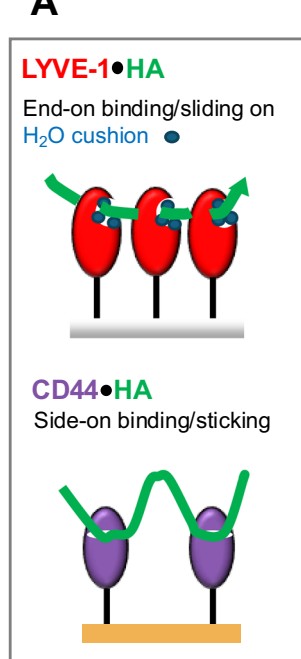

**A**

**LYVE-1•HA**

End-on binding/sliding on
H₂O cushion ●

**CD44•HA**

Side-on binding/sticking

**B**

**Interstitial fluid flow and chemokine - directed DC locomotion**

Collective HA unbinding from LYVE-1 and retention by CD44

DC

Lymphatic    endothelium

Transmigration to lumen of lymphatic capillary

**C**

**Dendritic cell CD44:HA glycocalyx with mix of free chain-ends and loops**

Bonds re-form post-separation (on CD44 side)

Dendritic cell

* HA *sticks* to CD44

⇑ Retractive force

Dendritic cell

⇑ Retractive force

Dendritic cell

⇑ Retractive force

Dendritic cell

Lymphatic endothelium    Lymphatic endothelium    Lymphatic endothelium    Lymphatic endothelium

* HA *slides* on LYVE-1

✗ Stochastic bond rupture at loops
↘ Sliding motion and escape through chain end

**Fig. 7 | Hypothetical model of how a sliding mode of HA adhesion/de-adhesion to LYVE-1 facilitates dendritic cell attachment and entry to lymphatic capillaries.** Individual panels show cartoon representations of: **A** the sequential engagement of an individual HA chain (green) by neighbouring LYVE-1 molecules (red) via end-on binding as opposed to engagement with CD44 via side-on binding, as determined by DFS. The water cushion (blue dots) in the LYVE-1 HABD observed by crystallography and calculated by MD is predicted to facilitate HA chain sliding, **B** a migrating DC adhering to the basolateral endothelium of an initial lymphatic capillary via its HA glycocalyx and transmigrating at a junction via sequential binding/unbinding of HA from LYVE-1 at the DC uropod, and **C** a more detailed model showing the proposed sequence of events during such unbinding of the DC HA glycocalyx from lymphatic endothelium (boxed area in **B**) by collective reverse sliding of HA through LYVE-1, together with stochastic bond rupture between HA and CD44 to leave the DC glycocalyx intact. For purposes of clarity, LYVE-1 is depicted as a monomer.

microscopy[3,8,56,57], as well as critical for the HA-dependent interactions between DCs and T cells that support immune synapse formation during antigen-specific T cell responses in downstream lymph nodes[17]. Indeed, when one considers the low shear stress (<5 μN/cm²) environment that surrounds lymphatic capillaries in vivo[3,5], this sliding mode of LYVE-1•HA interaction[16,57] seems optimally suited to the integrin-independent crawling and squeezing of DCs that mediates their transit through the LYVE-1 lined button junctions to the capillary lumen[10,12,31,32,58,59]. This contrasts with the high shear environment of post-capillary blood flow (up to 0.1 mN/cm²)[60] in which CD44 mediates

the capture of circulating lymphocytes and neutrophils (which lack an HA coat) through its more conventional sticking interactions with HA in the glycocalyx of vascular endothelium[18–20].

Finally, while our present study focussed on monomeric LYVE-1, it should be noted that the receptor can also form disulfide-linked homodimers in vivo, an association that increases its apparent HA binding affinity by ~15-fold[26,36]. How the pairing of HABDs might generate such avidity remains to be determined, as our attempts to crystallise appropriate LYVE-1 constructs spanning the juxtamembrane domain with its critical disulfide-forming cysteine residue (C197

mLYVE-1 / C201 hLYVE-1)[26] have been unsuccessful, owing to the unstructured nature of this heavily O-glycosylated region. Nevertheless, as we showed in our DFS assays, the same unusual HA unbinding mechanics applied to both the monomeric and dimeric forms of the receptor, affirming the sliding interaction is an intrinsic property of the HA-binding Link domain in LYVE-1, independent of receptor self-association state.

In conclusion, our studies identify LYVE-1 as an HA receptor whose preference for a particular topology (open chain ends), combined with a distinct mode of sugar-binding/unbinding and density-dependent 'superselectivity'[61,62] help explain its evolution as a key regulator of lymph vessel entry[8,24]. Ongoing studies should reveal how these distinct properties can be exploited for the development of LYVE-1-based therapies that block unwanted immune and inflammatory responses by disrupting lymphatic trafficking.

# Methods

## Cell lines
Hamster CHO-K1 cells were maintained in adherent cell culture in T75 flasks incubated at 37 °C, 5% CO$_2$ in DMEM (Thermo Fisher # 10938-025), supplemented with 10% (v/v) foetal bovine serum (Sigma # F-9665), 1% (v/v) Penicillin/Streptomycin (PS, Thermo Fisher PS, Thermo Fisher # 15070063; final concentrations 50 U/ml Penicillin, 50 µg/ml Streptomycin), 2 mM L-glutamine (Sigma # G7513-100ML) and 1 mM sodium pyruvate (Sigma # S8636-100ML). Human HEK 293T cells were grown in adherent cell culture in T75 flasks incubated at 37 °C, 5% CO$_2$ in DMEM, supplemented with 10% (v/v) FBS, 1% (v/v) PS and 2 mM L-glutamine.

## Cell line authentication
HEK 293T (ATCC # CRL-3216) used in the preparation of virus-like particles for subsequent transduction of CHO-K1 were originally obtained from the Cancer Research UK (CRUK) repository under a reagent provision programme.

CHO-K1 (D28-W1) line was obtained from Lonza Biologics (formerly Celltech Ltd) in 1988 and for which we no longer hold records.

## Generation of mouse and human LYVE-1 HABDs for crystallography
To determine the basis for the HA-binding properties of LYVE-1 at an atomic level, we generated recombinant soluble versions of the N-terminal HA-binding domains for expression in CHO cells. In order to identify optimal points for truncation in each case, we referred to our previous crystallography of CD44, that defined an extension of the classical Link module with one additional N-terminal (β0) and two additional C-terminal β strands (β8,9). We initially generated an equivalent-sized mouse LYVE-1 construct but this consistently failed to yield a functional, folded polypeptide. As the C-terminal region spanning the predicted β8 and β9 strands has limited homology with CD44 and contains a high proportion of serine and threonine residues (Supplementary Fig. 1), we considered the likelihood that it forms the distal portion of a heavily O-glycosylated membrane-proximal stalk rather than part of the globular LYVE-1 HABD which therefore might be more compact than previously anticipated. Accordingly, we generated truncated His$_6$ tagged constructs for the mLYVE-1 and hLYVE-1 HA-binding domains that terminated at the end of, or just beyond β7 (Supplementary Fig. 1) by means of PCR with Pfu Ultra AD polymerase (Agilent # 600385) and appropriate cDNA templates using the primers mLY-14 BamHI forward (5′CGCGGATCCGGAGGGATCTGCACAATGC TCC-3′) with either mLY429 6His*XhoI reverse (5′GCGCTCGAGTTAat-gatgatgatgatgatgAACGATTTCTGGAATGCAGGAG-3′) or mLY423 6His*XhoI reverse (5′GCGCTCGAGTTAatgatgatgatgatgatgTTCTGGAA TGCAGGAGTTAACCC-3′), and hLY-14 BamHI forward (5′CGC GGATCCGAAGGGTAGGCACGATGGCC-3′) with hLY432 6His*XhoI reverse (5′GCGCTCGAGTTAatgatgatgatgatgatgGATAATTTCTGGAAT

GCACGAG-3′) to yield respectively mLYVE-1 Δ141, mLYVE-1 Δ143 and hLYVE-1 Δ144. Sequences for the polyhistidine tags are shown in lower case and for stop codons in bold.

The products were digested with BamHI and XhoI (both from NEB # R3136S and # R0146S) and cloned into a variant of the pHR SIN vector[63] carrying an internal ribosome entry site (IRES) upstream of emerald fluorescent protein. Following the transformation of E.coli DH5α (Thermo Fisher # 18263012), error-free clones were selected and propagated, prior to transient transfection of 70% confluent HEK 293T cells along with the packaging plasmids pMD.G and p8.91 using Genejuice (Merck # 70967-6). After 48–72 h in culture, packaged lentiviruses were harvested from the HEK 293T cell supernatants and passed through a 0.45-µm filter, before transduction of CHO-K1 cells, which were then incubated overnight before the supernatants were replaced with fresh growth medium. After a further 48 h, emerald-expressing cells were selected by fluorescence-activated cell sorting (FACS) and maintained as lines for the production of soluble LYVE-1 His$_6$ recombinant protein.

For large-scale protein purification, CHO-K1 lines expressing the appropriate LYVE-1 HA-binding domains were then expanded at 37 °C, 5% CO$_2$ in high-glucose DMEM supplemented with 10% FCS, 1 mM sodium pyruvate, 1× penicillin/streptomycin. In cases where the protein was required without N-glycan modification, culture media were further supplemented with the class I α-mannosidase inhibitor, kifunensine (Bio Techne # 3207). On reaching confluence in 15-cm tissue culture petri dishes, the media were replaced by low serum medium (2% FCS, otherwise, as described above). After a further 4 days, tissue culture supernatants were aspirated and passed through a 0.22-µm filter before diluting 1:3 in PBS. Imidazole, pH 7.4, and NaCl were added to give final concentrations of 20 and 300 mM, respectively. His-tagged proteins were extracted by passing the diluted and supplemented supernatants over 5 ml His Trap columns (Cytiva # 17525501) before washing with PBS, supplemented with 20 mM imidazole (Sigma # 5674), and additional NaCl to a final concentration of 300 mM) and then eluting (PBS, 500 mM imidazole, 300 mM NaCl, final concentration). After elution, proteins were buffer-exchanged into PBS and concentrated before size exclusion. For CHO-K1 cell lines that had been grown in the presence of kifunensine, the His Trap column eluates were instead buffer-exchanged into 50 mM sodium acetate pH 6.0, 150 mM NaCl prior to deglycosylation (removal of N-linked glycans except for the initial N-acetyl glucosamine) with Endo Hf (NEB # P0703S) at 20 °C for 16 h, prior to final purification by size exclusion chromatography.

## Protein crystallisation and structure determination
Crystallisation screening was carried out by sitting-drop vapour-diffusion methods in CrystalQuick 96-well plates (Greiner # 609120) by mixing 100 nl protein solution with 100 nl reservoir and equilibrating against 95 µl of reservoir at 20 °C. mLYVE-1 was crystallised from a stock solution at 17.6 mg/ml, in 25% w/v polyethylene glycol 3350, 100 mM bis-Tris pH 6.5, and hLYVE-1 was crystallised from a stock solution at 36 mg/ml, in 0.1 M sodium/potassium phosphate pH 5.5, 2.5 M NaCl. mLYVE-1•HA8 was crystallised from a solution at 18 mg/ml in 20% polyethylene glycol 5000, 100 mM bis-Tris pH 6.5 and hLYVE-1•HA10 was crystallised from a solution at 27.6 mg/ml, in 20% w/v polyethylene glycol 3350, 100 mM bis-Tris propane pH 6.5 and 200 mM sodium malonate.

The crystal diffraction data were collected on beamlines I24 (mLYVE-1 apoprotein, mLYVE-1•HA8 and hLYVE-1 apoprotein) at a wavelength of 0.9686 Å and I04-1 (hLYVE-1•HA10) at a wavelength of 0.9159 Å at Diamond Light Source (Didcot, UK).

The crystallographic data were indexed and scaled using autoPROC[64] or Xia2[65] software. The structure of hLYVE-1 was determined in an unliganded state by single-wavelength anomalous dispersion, using phenix.autosol[66], for a dataset that was collected from a

single crystal with a spacegroup of P42₁2, at a wavelength of 1.77 Å and extending to 1.91 Å resolution. Model building was conducted using phenix.autobuild[67] prior to remodelling and refinement using Coot[68] and phenix.refine[69]. The structures of the apo forms of hYLVE1 for a new dataset from a second crystal grown in the same conditions (collected at a wavelength of 0.9686 Å), and of mLYVE-1, were subsequently determined by molecular replacement using Phaser within Phenix[69,70] and with the experimentally-phased hYLVE1 apo structure as a search model. The mLYVE-1 apo structure (residues 29–143 built) was determined at a resolution of 1.54 Å, and the final hLYVE apo structure (residues 24–144) at a resolution of 1.64 Å. Subsequently, structures for mLYVE•HA8 and for hLYVE-1•HA10 were determined using molecular replacement with their respective apo state models, again using Phaser within Phenix[69,70], prior to model rebuilding and refinement as for the apo states. The data collection and structure refinement statistics for each of the reported structures are summarised in Supplementary Table 1, and both $2F_o$-$F_c$ (apo and HA-bound) and composite omit maps of the structures determined in Phenix[69,70] are shown in Supplementary Fig. 12.

The buried surface area of binding was calculated using Areaimol[71]. Structural figures were drawn using Pymol (The PyMOL Molecular Graphics System, Version 2.0 Schrödinger, LLC) and Chimera[72].

## Generation of LYVE-1 and CD44 domains for dynamic force spectroscopy

For the soluble wild-type hLYVE-1 His₁₀ tag full ectodomain, the sequence up to and including residue number 238 (LYVE-1 Δ238 His₁₀) was amplified from human *lyve1* cDNA using the high fidelity polymerase *Pfu* Ultra AD with the primers: hLY-14 *Hind*III forward (5′-GCGAAGCTTGAAGGGGTAGGCACGATGGCCAGGTG-3′) and hLY+714 10His*XbaI reverse (5′-CGCTCTAGA**TTA**atggtgatggtgatggtgatggtgatggtgCGTGGGGACACCTCCAAACCC-3′), where the His₁₀ tag (reverse primer) is shown in lower case, stop codons are italicised in bold and restriction enzyme recognition sites are underlined, respectively.

The PCR product was digested with *Hind*III and *Xba*I (NEB # R3104S and # R3133S), ligated into the expression vector pEE14 and transformed into *E.coli* DH5α (Thermo Fisher). An error-free clone was selected and transfected into Chinese hamster ovary (CHO) K1 cells using the Fugene lipid transfection reagent (Promega # E2691). LYVE-1 expressing cells were then selected in media containing 40 µM methionine sulfoximine and a high expression clone identified by Western blotting of culture supernatants with the hLYVE-1 specific monoclonal antibody 8C[38].

For large-scale production of the recombinant LYVE-1 ectodomains, appropriately transfected CHO-K1 cells were grown in 15 cm culture dishes and expanded at 37 °C, 5% CO₂ in high-glucose Dulbecco's modified Eagle's medium (DMEM) supplemented with 10% foetal calf serum that had been dialysed against PBS (First Link # 02-15-850), pH 7.5, 1 mM sodium pyruvate, 1× GS Medium Supplement (Merck # GSS-1016-C), 1× penicillin/streptomycin and 40 µM L-methionine sulfoximine (Sigma # GSS-1015-F). On reaching confluence, the medium was replaced by serum-free DMEM supplemented with 1 mM sodium pyruvate, 2 mM sodium butyrate (Sigma # B5887) and 1× penicillin/streptomycin, and cells cultured for a further 4 days before aspiration of the supernatant, which was passed over a 0.22-µm filter before dilution 1:3 in PBS, supplemented with imidazole and NaCl to final concentrations of 20 and 300 mM, respectively. The His-tagged protein was then extracted by passing the diluted and supplemented supernatant over a 5 ml His Trap column, which was washed with PBS, pH 7.5 containing 20 mM imidazole and 300 mM NaCl and then eluted with PBS containing 500 mM imidazole and 300 mM NaCl. The purified wild-type hLYVE-1 Δ238 His₁₀ protein was finally buffer-exchanged into PBS and concentrated prior to size exclusion chromatography on a Superdex Increase 200 10/300GL (Cytiva # 28990944) column to remove aggregates, yielding a mixture of monomer and dimer at an approximate ratio of 3:1.

To generate the soluble, non-dimerising hLYVE-1 Cys201Ala His₁₀ tag mutant ectodomain, an amplicon was produced using a mutated version of the *lyve1* cDNA carrying the Cys201Ala mutation as template and the primers hLY-14 *Bam*HI forward (5′-GCGGGATCCGAAGGGGTAGGCACGATGGCCAGGTG-3′) and hLY+714 10His*XhoI reverse (5′-GCGCCTCGAG**TTA**atggtgatggtgatggtgatggtgCGTGGGGACACCTCCAAACCC-3′) to yield hLYVE-1 C201A Δ238 His₁₀. Similarly, to generate the wild-type hLYVE-1 BirA (for biotinylation) and His₁₀ dual tag ectodomain, wild-type *lyve1* cDNA served as template in the production of an amplicon using the primers hLY-14 *Bam*HI forward and hLY+714 BirA 10His*XhoI reverse (5′-GCGCCTCGAG**TTA**atggtgatggtgatggtgatggtgatggtgccactcgattttctgtgcctcgaagatgtcattcaggccCGTGGGGACACCTCCAAACCC-3′). Notation is as described previously, except the sequence encoding the motif (GLNDITEAQKIEW) for biotin modification via BirA ligase is underlined and lowercase.

In the description of the cloning and expression of such constructs below, this tag is referred to as the BirA recognition sequence (BirA RS).

The PCR products were digested with *Bam*HI and *Xho*I and cloned into a variant of the pHR SIN vector[63] carrying an internal ribosome entry site (IRES) upstream of emerald fluorescent protein prior to packaging and production of lentivirus and transduction of CHO-K1 cells as described above for the truncated LYVE-1 HA binding domain constructs. The hLYVE-1 Cys201Ala Δ238 BirA RS His₁₀ and wild-type hLYVE-1 Δ238 BirA His₁₀ dual tag proteins were then extracted from culture supernatants and purified by His-tag affinity and size exclusion chromatography using the same procedure as that for the wild-type ectodomain[30].

The BirA tag in wild-type hLYVE-1 Δ238 BirA RS His₁₀ was biotinylated (1 h room temperature) according to the protocol supplied with the BirA ligase kit (Avidity LLC # BirA500): 0.23 mg (10 nmol) hLYVE-1 Δ238 BirA RS His₁₀ in 90 µL of 50 mM Tris-HCl, pH 8, was combined with 10 µL 10X supermix and 1 µg BirA enzyme in a total reaction volume of 100 µL. The resulting protein (hLYVE-1 Δ238 with a biotin/His₁₀ dual tag) was then re-purified by size exclusion chromatography.

To generate the soluble hCD44 ectodomain hCD44 Δ267 biotin/His₁₀, the hCD44 Δ267 BirA RS/His₁₀ protein was produced recombinantly in CHO-K1 cells, purified by His-tag affinity and size exclusion chromatography, then biotinylated as described in detail previously[30] (the biotin tag was not deployed in the present study).

## Hyaluronan preparations

Hyaluronan polymers (HA) with specific molecular masses and quasi-monodisperse size distributions were purchased from Hyalose: HA with biotin at its reducing end had a molecular mass of 840 ± 60 kDa (HA-b 840, Hyalose # Select-HA B1000); non-derivatised HA used for binding assays and blocking assays had molecular masses of 250 ± 12 kDa and 58 ± 3 kDa (both Hyalose # Select-HA 250 and # Select-HA 50), respectively. Lyophilised HA was dissolved and gently shaken for 2 h in ultrapure water to make a stock at 1 mg/ml.

An HA derivative with multiple biotins grafted along the polymer chain (HA-*g*-b) was prepared from high molecular weight research grade HA (Lifecore # HA15M-5) with an estimated molecular mass ranging from 1.0 to 1.8 MDa using a procedure modified from that detailed previously[73]. Briefly, 5 mg HA (equivalent to 13 µmol of disaccharides) and desired quantities of 1-ethyl-3-(3-dimethylaminopropyl)carbodiimide (EDC, Thermo Fisher # 22980)) were incubated in 1 ml of 0.1 M MES, pH 5.5. Biotin-LC-hydrazide (Thermo Fisher # 21340) in DMSO was then added to a final concentration of 1 mM, and the mixture was stirred overnight (25 °C) before dialysis against ultrapure water. The EDC quantities used here were 8.1 µg (0.042 µmol) or

40.7 µg (0.21 µmol). The nominal degree of biotinylation (i.e., assuming complete reaction) would thus be 0.3 or 1.6% of the HA disaccharides, corresponding to average HA contour lengths between biotinylation sites of approximately 300 nm or 60 nm, respectively. The actual degree of biotinylation is likely to be lower, and the average length of HA stretches between biotinylation sites is thus larger.

A pair of quasi-monodisperse 320-kDa size-matched HA constructs, one with a biotin at the reducing end (HA-b 320) and the other with a biotin at the non-reducing end (b-HA 320), were prepared by two different chemoenzymatic schemes. The first construct, HA-b 320, was synthesised via synchronised, stoichiometrically controlled polymerisation of UDP-GlcNAc and UDP-GlcA (Sigma # U4375 and # U6751) donors with a biotinylated HA tetrasaccharide primer utilising purified *Pasteurella multocida* HA synthase (PmHAS) enzyme[74]. In brief, the polymer product was purified by extraction, dialysis and ethanol precipitation, then quantitated by the carbazole assay. Coupled size exclusion chromatography multi-angle light scattering (SEC-MALS[74]) gave an average $M_W$ of 317 kDa, and a dispersity $M_w/M_n = 1.03$.

The other construct, b-HA 320, was synthesised in a multi-step process that involved making the HA chain[74], adding a functional amino group, and then installing the biotin at the non-reducing terminus. First, synchronised polymerisation with a HA tetrasaccharide primer was used to create the HA polysaccharide chain; this purified polymer gave an average $M_W$ of 321 kDa, and a dispersity $M_w/M_n = 1.002$, as determined by SEC-MALS. Second, to assure that all HA chains had the identical terminal sugar, a single sugar addition was performed with 5 equivalents of UDP-GlcA (50 mM Hepes, pH 7.2, 1 mM MnCl₂, 2 mg/ml polymer, 1 mg/ml PmHAS, overnight reaction at 22 °C). This GlcA-capped HA precursor (GlcA-HA) was isolated by preparative descending paper chromatography (65:35 ethanol, 1 M ammonium acetate, pH 5.5, development solvent with Whatman 3MM); the polysaccharide remains at the origin of the paper strip, whereas any excess UDP-sugars and the UDP by-product migrate substantially down the paper. The origin with the target was cut out, air-dried and then eluted with water. The samples were then frozen, and the volatile salt was removed by three cycles of lyophilisation from water (typical overall yield, >95%). Third, an artificial donor with a protected amine, UDP-GlcN[TFA] (5 equivalents; TFA, trifluoroactic acid; kindly provided by Robert Linhardt, Rensselaer Polytechnic Institute, Troy, NY, USA), was employed in another single sugar addition step in a fashion similar to work with heparosan-based polysaccharides[75], except PmHAS was the catalyst using the reaction conditions described above for GlcA addition at 30 °C. This GlcN[TFA]-HA polymer was then purified as for GlcA-HA except for an extraction with *n*-butanol prior to the preparative descending paper chromatography step. Fourth, the TFA group was removed by mild base treatment (MeOH/H₂O/triethylamine 2:2:1, 1 day at 22 °C) to expose the amino group. The reaction mixture was diluted twofold in water, frozen, and lyophilised to yield GlcN-HA. Fifth and finally, the unmasked terminal amine was then coupled to biotin using 200 equivalents of sulfo-NHS-LC-biotin (Thermo Fisher # 21327) in 0.1 M sodium phosphate, pH 7.0, 22 °C for 4 h. The b-HA product was purified by paper chromatography and quantitated by the carbazole assay. The HA-b and b-HA migrated as narrow bands with very similar sizes on agarose gels[74].

HA octasaccharides (HA8) and decasaccharides (HA10) used for crystallography and surface plasmon resonance analyses were purchased from Iduron (# HA08 and # HA10, respectively). All HA, LYVE-1 and CD44 preparations were either diluted in 10 mM HEPES (Sigma) pH 7.4 containing 150 mM NaCl before use, or stored undiluted in aliquots at −20 °C, and then used within a few weeks of thawing.

## Dynamic force spectroscopy
DFS was based on AFM, with functionalised AFM probes and planar substrates (see below). All AFM experiments were performed with NanoWizard II and NanoWizard IV systems in a working buffer at ambient conditions. Force curves were acquired at approach and retract velocities of typically 1 µm·s⁻¹; where desired, velocities were tuned between 0.5 and 12 µm·s⁻¹. The maximal applied load was 600 pN and the surface dwell time at maximal load was kept to a mimimum (i.e., 0 ms), except for experiments with HA in loop configuration, where dwell times between 2 and 3 s were used. For a given set of AFM probe, surface, and interaction settings, 50 force curves were acquired if the interactions did not exhibit stochastic variations (i.e., for LYVE-1 on HA in tail configuration) and several hundred to thousand individual force curves were acquired if the interactions exhibited stochastic variations (i.e., for CD44 on HA in tail configuration, and for LYVE-1 on HA in loop configuration). All experiments were performed at least twice with distinct yet identically prepared AFM probes and planar substrates.

Force curves were analysed with JPK Data Processing software (version 6.3.36). Force vs. distance curves exhibiting stochastic variations were analysed with the worm-like chain (WLC) model[76]

$$F = \frac{k_B T}{L_p} \left[ \frac{1}{4} \left( 1 - \frac{z}{L_c} \right)^{-2} + \frac{z}{L_c} - \frac{1}{4} \right] \qquad (1)$$

where $k_B T$ is the thermal energy and the probe-substrate distance $z$ was assumed to be equivalent to the polymer extension. The persistence length $L_p$ and the contour length $L_c$ were kept as adjustable parameters, unless otherwise stated. Only rupture events appearing at tip-substrate distances larger than 250 nm were considered for further analysis, to avoid bias by non-specific tip-sample interactions and to ascertain that $L_c \gg L_p$ as required for the validity of the WLC model. From Eq. (1), it can be seen that two chains of the same persistence and contour lengths, when stretched in parallel, would give rise to a WLC response with an effective persistence length of $L_{p,eff} = L_p/2$. Instantaneous loading rates $r$ were calculated from the effective spring constant $k_{eff}$, corresponding to the slope of the WLC fit close to the rupture, and the retract velocity $v$ as $r = k_{eff} v$. OriginPro 2019b software (version 9.6.5.169) was employed to determine means and standard deviations through non-linear regression analysis of histograms.

To prepare AFM planar substrates, appropriate soluble LYVE-1 or CD44 ectodomains were incubated for 30 min at typical concentrations of 5 µg/ml for LYVE-1 or 6.5 µg/ml for CD44 in the case of His-tag anchorage to PEG and SLB coatings, and 10 µg/ml for LYVE-1 in the case of biotin anchorage to SAv coatings (see below). These conditions lead to relatively dense receptor monolayers (root mean square (rms) distances $d_{rms}$ between 10 and 20 nm). Where desired, the grafting density was tuned by varying the incubation concentration between 0 and 10 µg/ml (see Supplementary Figs. 4, 7).

For AFM probes, gold-coated versions with a nominal cantilever spring constant of 6 or 30 pN/nm and a nominal tip apex diameter of 30 nm (BioLevers), and with a nominal spring constant of 60 pN/nm and a nominal tip apex diameter of 20 nm (NPG), were purchased from Bruker AFM Probes. The real cantilever spring constants were determined by the thermal noise method.

For OEG-functionalisation of AFM probes, oligo(ethylene glycol) (OEG) constructs were purchased from Polypure, one made of EG₇ with a hydroxyl group on one end and a thiol on the other ((OEG)₇-SH, Polypure # 10156_0795), and the other containing EG₁₀ with biotin on one end and a thiol on the other (b-(OEG)₁₀-SH, Polypure # 41156-1095). AFM probes were exposed to UV/ozone for 30 min, and then immersed overnight at 4 °C in an ethanolic solution of (OEG)₇-SH and b-(OEG)₁₀-SH at a total concentration of 1 mM and a molar ratio of 98:2. Prior to use, the functionalised probes were rinsed with ethanol, gently blow-dried with nitrogen gas and then immersed in working buffer.

For anchorage of HA to AFM probes, HA-b, b-HA and HA-*g*-b were attached to AFM probes exploiting the strong yet specific interactions

between SAv and biotin. To this end, AFM probes with a biotin-displaying OEG monolayer were first incubated with SAv. After rinsing, the cantilevers were then immersed in a solution of biotinylated HA. The AFM probes were kept wet at all times during this preparation process and their subsequent use.

For single-end grafting of HA chains (i.e., to generate exclusively HA tails), SAv was incubated for 20 min at 20 µg/ml, and then HA-b 840, HA-b 320 or b-HA 320 for 6 min at 2 µg/ml. We have previously shown that this leads to a dense monolayer of SAv and, for b-HA 840, to a sparse monolayer of HA. Specifically, the root mean square (rms) distance between HA-b 840 anchor points on the AFM tip was estimated to be at least 76 nm[30,77]. Considering the large radius of gyration ($R_g$) of the HA polymer (~75 nm[78]) and the small radius of the AFM tip apex (≤30 nm), it is thus expected that only one or at most a few HA chains are able to contact the receptor-coated substrate simultaneously. HA-b 320 and b-HA 320 were not previously reported, and their binding to SAv monolayers is characterised in more detail in Supplementary Fig. 11.

To anchor HA via multiple points along the chain (i.e., to generate HA loops), SAv was incubated for 20 min at 20 µg/ml, and then HA-$g$-b for 6 min at 2 µg/ml. Considering the geometry of the AFM probe and that SAv binding is mass-transport limited at low surface coverage, we estimate an rms distance for SAv of 13 nm under these incubation conditions[77]. This SAv coverage combined with sparse biotin grafting on HA-$g$-b yielded loops with contour lengths of several 100 nm as desired for DFS analysis.

### Estimation of HA: receptor binding rates from DFS data

In the case of LYVE-1 binding via the HA non-reducing end, DFS data consistently revealed a deterministic response indicating the engagement of many receptors with sugar during the limited time of the HA-coated AFM tip being in proximity of the receptor-coated surface. We consider that an open HA end can contact receptors once the tip-surface distance $z$ is comparable to the size of the HA random coil ($z \approx 2R_g$). The effective molar concentration of open ends is approximated as $c_{end} \approx (\frac{4}{3}\pi R_g^3 N_A)^{-1}$. The number of receptors within reach of an HA chain is $N_R \approx \pi R_g^2/d_{rms,R}^2$, with $d_{rms,R}$ the average (root mean square) spacing of receptors. The time to approach from $z = 2R_g$ to contact ($z = 0$) is $t_b = 2R_g/v$, with $v$, the approach velocity. From these estimates, we derive a lower bound for the binding constant as $k_{on} \gtrsim (N_R c_{end} t_b)^{-1} \approx \frac{2}{3}d_{rms}^2 N_A v$. With $d_{rms} \approx 20$ nm and $v = 1$ µm/s, we obtain $k_{on} > 10^5$ M$^{-1}$s$^{-1}$.

We note that the above-derived estimation is conservative, and that the exact binding rate constant likely is substantially higher. Approach curves for end-on binding of HA to LYVE-1 consistently revealed attractive interactions at distances of a few 100 nm (cf. Figs. 1C, 2E, G and Supplementary Figs. 5B, 10B). These distances are comparable, if not larger, than the average extension of an HA random coil, suggesting that the time of thermally driven transient extensions of an HA chain beyond its average size is sufficient for binding events to occur. These times are expected to be much shorter than $t_b$.

In the case of LYVE-1 binding to HA loops, DFS data demonstrated stochastic binding (Fig. 2A–C). To calculate $k_{on}$, the frequency of observing a bond was correlated with the local concentrations of binding sites on HA (defined by the size of the HA loop) and LYVE-1 on the surface, and the dwell time of the AFM tip at the surface. Details of the analysis are provided in Supplementary Fig. 8.

### Lipids for DFS and QCM-D

The 1,2-dioleoyl-$sn$-glycero-3-phosphocholine (DOPC) was purchased from Avanti Polar Lipids (# 850375 P). Tris-NTA-functionalised lipid analogues ((NTA)$_3$-SOA) were prepared as described earlier[79] and kindly provided by Changjiang You and Jacob Piehler (Osnabrück University, Germany). Small unilamellar vesicles (SUVs) in a working buffer were prepared by sonication from a mixture of DOPC and (NTA)$_3$-SOA (95:5 molar ratio), as described previously[80]. SUVs at a stock concentration of 1 or 2 mg/ml were stored at 4 °C under argon and used within 3 weeks.

### Quartz crystal microbalance with dissipation monitoring (QCM-D)

QCM-D was employed to validate the proper immobilisation of hyaluronan and its receptors. This technique measures the changes in resonance frequency, $\Delta f$, and dissipation, $\Delta D$, of a sensor crystal upon molecular adsorption on its surface. The QCM-D response is sensitive to the areal mass density (including hydrodynamically coupled water) and the mechanical properties of the surface-bound layer. To a first approximation, a decrease in frequency ($\Delta f$) corresponds to increased mass, while a low (high) response in dissipation ($\Delta D$) corresponds to a rigid (soft) film.

Measurements were carried out with a Q-Sense E4 system equipped with Flow Modules with flow rates of 5 to 20 µl/min at a working temperature of 23 °C. $\Delta f$ and $\Delta D$ were collected at six overtones ($i = 3$, 5, 7, 9, 11, 13). Changes in dissipation, $\Delta D$, and normalised frequency, $\Delta F = \Delta f_i/i$, for $i = 3$ are presented. All other overtones provided similar information. All experiments were carried out in duplicate. Data were analysed using OriginPro 2019b (version 9.6.5.169) software.

### Preparation and functionalisation of planar substrates

QCM-D sensors with a silica coating (# QSX303), with a gold coating (# QSX301), and with a His-tag-capturing coating (# QSX340) were obtained from Biolin Scientific. Silicon wafers (9 mm × 9 mm) with a native oxide layer of approximately 2 nm were from University Wafers (# 6985).

Silica-coated substrates were cleaned with 2% ($w/v$) sodium dodecyl sulfate (SDS, Sigma # 75746) for 30 min, rinsed thoroughly with ultrapure water followed by blow-drying with nitrogen gas, and treated with UV/ozone for 30 min prior to each measurement. Gold-coated sensors were used as received and treated with UV/ozone for 30 min prior to use. His-tag-capturing sensors were used as received and regenerated with 500 mM imidazole (25 min) followed by 5 mM CuSO$_4$ (15 min) for repeated functionalisation.

For anchorage of CD44 and LYVE-1, in most cases and unless otherwise stated, His-tagged protein constructs were directly immobilised on His-tag-capturing QCM-D sensors, for both QCM-D and DFS analyses. This sensor surface has a passivating layer of poly(ethylene glycol) (PEG) and bares divalent metal ions for anchoring His-tagged molecules. A QCM-D characterisation of LYVE-1 (i.e., hLYVE-1 Δ238 with either a His$_{10}$ single tag or a biotin/His$_{10}$ dual tag) immobilisation on His-tag-capturing PEG substrates is shown in Supplementary Fig. 18; a QCM-D characterisation of CD44 immobilisation has been presented previously[30] and is not explicitly shown here.

In one instance (Supplementary Fig. 5), a fluid-supported lipid bilayer (SLBs) exposing Ni$^{2+}$-loaded tris-NTA moieties was used to anchor hLYVE-1 Δ238 with a His$_{10}$ single tag for DFS analyses. SLBs were formed on silica-coated surfaces from SUVs by the method of vesicle spreading, as described previously[81]. A complementary QCM-D characterisation of LYVE-1 immobilisation on His-tag-capturing SLBs is shown in Supplementary Fig. 19.

In another instance (Fig. 2D–G), a complementary biotin tag in hLYVE-1 Δ238 with a biotin/His$_{10}$ dual tag was used to immobilise the receptor for DFS analyses. To this end, gold-coated surfaces were first functionalised with a SAv monolayer, as described for OEG-functionalisation of AFM probes (see Dynamic force spectroscopy section above), and then incubated with LYVE-1. SPR analysis of LYVE-1•HA binding affinity was also performed with biotin-anchored LYVE-1, although in this case, a bespoke methyldextran matrix with covalently coupled SAv was used (see Supplementary Fig. 9 for details).

Anchorage via the His-tag or biotin tag affords the presentation of the receptor ectodomains at a defined orientation faithful to the

situation on the cell membrane. The presentation on PEG and SLBs (for His tags) and SAv (for biotin tags) is expected to be similar[30], except for lateral mobility, which is preserved on the fluid SLBs but impaired on PEG and SAv (see Supplementary Figs. 18–20).

## Molecular dynamics simulations

The mLYVE-1/hLYVE-1 crystal structures were solved in this work, while the mCD44 crystal structures were taken from the Protein Data Bank (2JCP (CD44 apoprotein) and 2JCR (CD44/HA complex)). The N-terminus of mLYVE-1, hLYVE-1 and mCD44 harbours the signal peptide (residues 1–23, 1–19 and 1–22, respectively, Uniprot ID: Q8BHC0, Q9Y5Y7 and P15379) which is cleaved off during maturation. Consequently, the N-terminus of mCD44 (starting from residue 23) was positively charged. In contrast, the mLYVE-1 and hLYVE-1 proteins used for crystallography lacked several residues beyond the signal peptide (residues 24–28 and 20–24, respectively), and hence the existing N-terminal residues were modified with a neutral acetyl cap, to avoid introducing a non-natural charge in comparison to the full-length protein. The most C-terminal residues of all receptors were modified with a neutral $N$-methyl amide cap for the same reason. Three disulfides were present in each protein (residues pairs 60–127, 35–138, 84–105 for mLYVE-1; 61–128, 36–139, 85–106 for hLYVE-1; 32–134, 57–123, 81–101 for mCD44.

To prepare for simulations, all glycosylations (up to three Asn-N-linked GlcNAc moieties per protein), water and co-solvent molecules were discarded. Hydrogens were added to the proteins and protein:HA complexes using the LEaP module of AMBER20[82], and their mass was repartitioned to 3 Da using the ParmEd module. All Asp, Glu, Arg, Lys, free N- and C-termini and D-glucuronic acid moieties were thus charged. The state of His residues was suggested by the Reduce programme[83] as follows: in mLYVE-1, His128 was flipped and protonated at Nε and in mCD44, His96 was flipped and protonated at Nδ and His118 was flipped and protonated at Nε. The proteins and protein:HA complexes were immersed in an octahedral box of water molecules (TIP3P or TIP4P/2005)[84,85] extending at least 12 Å from the solute, with van der Waals closeness parameter set to 0.7. Sodium and chloride ions were added to a concentration of 150 mM in each case to maintain overall neutrality. All the specifications of the systems can be found as well as the starting coordinates, force field topologies and simulation input files are provided via public repository[86].

With regard to models and parameters, the systems were described by standard and well-tested all-atom force fields ff14SB[87] for the proteins and GLYCAM06j-1[88] for HA. An explicit water model was needed to describe bridging water networks, although its choice may result in ligand unbinding (as was the case for TIP4Pew and OPC models) or rearrangement (TIP3P) in the case of CD44•HA[89]. Non-polarisable force fields were preferred because their use allowed us to achieve larger sampling compared to explicitly polarisable force fields. The timescale of a selected H-bond is shown in Supplementary Fig. 21, indicating that both states are sufficiently populated during MD. Given that non-polarisable force fields overestimate charged interactions[39], we used them cautiously as an extension of crystallographic observations. We note that the recently described implicitly polarisable force fields appear to be an ideal approach to describing the dynamic nature of interactions between hyaluronan and proteins[39].

For relaxation of the solvent, counterions and hydrogens on the solute, this achieved by minimisation (3000 cycles of steepest descent) and MD with time step of 1 fs, SHAKE restraints[90] on bonds with hydrogen atoms and a nonbonded cutoff of 9 Å. First, MD used the NVT (canonical) ensemble for 100 ps, then NpT (isothermal–isobaric) ensemble for 300 ps. The temperature was kept at 300 K using the Berendsen thermostat and the pressure was maintained at 1 atm using Berendsen barostat[91]. Restraints on the solute non-hydrogen atoms of 50 kcal.mol⁻¹ Å⁻² were gradually decreased in the next steps to 25 (10 ps), 10 (10 ps), 5 (50 ps), 2.5 (50 ps) and 1 (70 ps) kcal.mol⁻¹ Å⁻².

Thereafter, 300 ps of NpT MD with released protein side chains and HA ensued, i.e. the restraints of 500 kcal.mol⁻¹ Å⁻² were now only applied to the protein backbone. This was followed by 300 ps of unrestrained MD. A 1 μs MD production in NpT (300 K and 1 atm) was run in triplicates using the PMEMD programme of AMBER20[82]. The temperature was controlled using a Langevin thermostat with a collision frequency of 2 ps⁻¹. Thanks to the hydrogen mass repartitioning, we were able to increase the time step to 4 fs. Frames were saved every 100 ps.

For subsequent analysis of simulations, this were carried out using the CPPTRAJ module of AMBER20[82]. Trajectories were analysed for structural stability using root mean square deviation (RMSD) metrics. To focus on the crystallographic binding mode, 500-ns-long portions of trajectories in which it was retained were analysed. These are provided via public repository[86]. Due to the high flexibility of the protein termini, we restricted the analysis to residues 29 to 138, 24 to 133 and 25–134 for mLYVE-1, hLYVE-1 and mCD44, respectively. Prior to the root mean square fluctuation (RMSF) measurements, alignment over the non-hydrogen atoms of the above-mentioned residue ranges was performed. Hydrogen bonding was assessed using the following criteria: 3.6 Å cutoff for acceptor⋯donor distance and 120–180° range for acceptor⋯H-donor angle. We note that the timescale of the simulations was sufficient to sample H bonds occupancies (Supplementary Fig. 18). Bridging water molecule occupancies were calculated by summing up binary, ternary, and quaternary interactions (the cutoff for each was set to a minimum of 10%).

For restrained MD analyses, to visualise densities of water molecules bridging HA6 with mLYVE-1, hLYVE-1 or mCD44, we ran 10 ns NpT MD simulations with the settings as in the relaxation stage above, but with the protein and HA fixed with restraint of 1000 kcal.mol⁻¹ Å⁻². The densities for the 50 closest water sites in the HA binding groove were calculated via the VOLMAP tool in the CPPTRAJ module of AMBER20[82], utilising a scaling factor for oxygen radii of 1.36 and a buffer of 5 Å. The water densities and associated topologies and trajectories are provided via public repository[86].

## Reporting summary

Further information on research design is available in the Nature Portfolio Reporting Summary linked to this article.

## Data availability

The crystal structures generated in this study have been deposited in the RCSB Protein Data Bank database under accession codes 8ORX (mLYVE-1 Apo), 8OS2 (hLYVE-1 Apo), 8OX3 (mLYVE-1 with HA8), and 8OXD (mLYVE-1 with HA10). Plasmids generated in this study will be made available on request. The supporting computational data are provided on Zenodo https://zenodo.org/records/14802638[86]. All other unique/stable reagents generated in this study are available from the lead contact with a completed Materials Transfer Agreement. PDB codes of previously published structures used in this study are 2JCQ, 2JCR, and 2JCP. Source Data are provided as a Source Data file. Source data are provided with this paper.

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

## Acknowledgements

This work was financially supported by Unit and other grant funding from the UK Medical Research Council (MC_UU_00008/2 to D.G.J., MR/N000331/1 to R.J.C.G. and T.N.), grants from the European Research Council (ERC-2012-StG-306435) to R.P.R., the UK BBSRC (BB/R000174/1 to R.P.R. and E.P., BB/X00158X/1 to R.P.R. and BB/X007278/1 to R.P.R. and D.G.J.), the European Union Horizon 2020 research and innovation programme (Marie Sklodowska-Curie grant agreement 795605) to M.L., and a Health Research Grant from Oklahoma Center for the Advancement of Science and Technology to P.L.D. We acknowledge Diamond Light Source for access and support of the data collection facilities. Biacore binding experiments were supported by MRC funding MR/X013227/1 to I.M. The Division of Structural Biology was a part of the Wellcome Centre for Human Genetics, Wellcome Trust Core Grant Number 090532/Z/09/Z. F.B. thanks the Biochemical Imaging Center at Umeå University for providing access to AFM. M.L. gratefully acknowledges the computer resources and technical support provided by Cineca, University of Bologna, under Project HPC-EUROPA3 (INFRAIA-2016-1-730897), with the support of the EC Research Innovation Action under the H2020 Programme. We also thank Changjiang You and Jacob Piehler (Osnabrück University, Germany) for kindly providing tris-NTA-functionalised lipid analogues, and Robert Linhardt (Rensselaer Polytechnic Institute, Troy, NY, USA) for kindly providing UDP-GlcN[TFA].

## Author contributions

F.B., K.C., I.M., R.J.C.G., R.P.R., D.G.J. and S.B. designed the experiments. F.B., S.B., T.N., K.C. and I.M. obtained the experimental data. P.L.DeA and D.E.G. provided essential hyaluronan preparations. E.P. and M.L. designed and M.L. performed the molecular dynamics simulations. E.P. and R.P.R. developed the reductionist model and performed computer simulations. F.B., S.B., T.N., E.P., M.L., R.J.C.G., R.P.R. and D.G.J. analysed data and interpreted results. D.G.J., R.P.R. and R.J.C.G. conceptualised the project. D.G.J., R.P.R., R.J.C.G., P.L.DeA. and M.L. obtained funding. D.G.J., R.P.R., R.J.C. and M.L. drafted the manuscript. All authors participated in reviewing and editing the manuscript.

## Competing interests

The authors declare no competing interests.

## Additional information

¹School of Biomedical Sciences, Faculty of Biological Sciences, University of Leeds, Leeds LS2 9JT, UK. ²School of Physics and Astronomy, Faculty of Engineering and Physical Sciences, University of Leeds, Leeds LS2 9JT, UK. ³Bragg Centre for Materials Research, University of Leeds, Leeds LS2 9JT, UK. ⁴Astbury Centre for Structural Molecular Biology, University of Leeds, Leeds LS2 9JT, UK. ⁵CIC biomaGUNE, Paseo Miramon 182, 20014 Donostia-San Sebastián, Spain. ⁶Medical Research Council Translational Immune Discovery Unit, MRC Weatherall Institute of Molecular Medicine, University of Oxford, Oxford OX3 9DS, UK. ⁷Division of Structural Biology, Wellcome Trust Centre for Human Genetics, University of Oxford, Roosevelt Drive, Oxford OX3 7BN, UK. ⁸Department of Biochemistry and Physiology, University of Oklahoma Health Sciences Center, Oklahoma City, OK 73126, USA. ⁹Dipartimento di Fisica e Astronomia, Università di Bologna, 40127 Bologna, Italy. ¹⁰CERMAV, Université Grenoble Alpes, CNRS, 38000 Grenoble, France. ¹¹Institute of Organic Chemistry and Biochemistry, Czech Academy of Sciences, 16610, Prague 6, Czech Republic. ¹²Present address: Department of Clinical Microbiology, Umeå University, 90185 Umeå, Sweden. ¹³Present address: L1-60, Laboratory Block, University of Hong Kong, 21 Sassoon Road, Hong Kong, China. ¹⁴These authors contributed equally: Fouzia Bano, Suneale Banerji, Tao Ni. ✉e-mail: martin.lepsik@uochb.cas.cz; robert.gilbert@magd.ox.ac.uk; r.richter@leeds.ac.uk; david.jackson@imm.ox.ac.uk

