## [Transparent Peer Review file · Nature Communications]

Structure and unusual binding mechanism of the hyaluronan receptor LYVE-1 mediating leucocyte entry to lymphatics

Corresponding Author: Professor David Jackson

Version 0:

Reviewer comments:

Reviewer #1

(Remarks to the Author)

In this study, the authors reported the binding of LYVE-1 to HA and presented the first crystal structures of this binding interaction. Based on these structures and AFM data, they proposed a highly unusual sliding mode of ligand interaction. While the study is intriguing, some conclusions seem premature and possibly overstated.

The experimental details are somewhat difficult to follow. The results section should better explain how the LYVE-1 receptors and HA molecules were grafted onto the AFM tip and the surface. Additionally, the experimental methods contain unclear details. For example, the AFM planar substrates section mentions that LYVE-1 can be grafted using a histidine tag, but it is unclear how the NTA surface is prepared. Have QCM-D sensors been used?

The noise level observed in the AFM curves is surprisingly low. Theoretically, the noise level is directly related to thermal equilibrium and the spring constant, with the formula $F = (k_b * k_c * T)^{1/2}$. A quick calculation yields a theoretical noise level (with ideal instruments) of 5 to 15 pN. Some curves presented in the paper have noise levels well below these values (e.g., Figure 1c, first inset; Figure 2B). Could there be an issue with the calibration of the instrument or the cantilever?

The shape of the interaction curve between HA and LYVE-1 is unusual and sometimes seen during the desorption of certain polymers. If the experiments were conducted on QCM-D sensors with poorly characterized chemistry, could the authors repeat the experiments on a custom-made surface with well-defined chemistry, such as a gold-coated surface with NTA alkanethiols?

One puzzling aspect is that at the end of the rupture between HA and the surface, the final event should reflect the rupture of a single bond, similar to what is observed in Figure 2B. This should allow observation of a single rupture between an HA chain and a single receptor. This suggests that the unusual curves may be non-specific interactions. If this is incorrect, the authors need to provide very convincing experiments. For instance, they could create surfaces with varying NTA group densities to control receptor density or use point mutations in the LYVE-1 binding pocket to test specificity.

Regarding the DFS in Figure 2C, only four points were obtained within a very narrow loading rate range. To ensure accurate kinetics extraction, a broader loading rate range should be analyzed (at least over 2-3 log). As it stands, the significance of the Bell-Evans fit slope is questionable. A horizontal fit might also be plausible (meaning non-specific interaction). Could the authors explore a larger range or provide a statistical analysis to evaluate the fit's error and the slope's significance?

The discussion section requires careful revision, as some points are not supported by data. For instance, lines 339-340 state, "our DFS analyses revealed that LYVE-1 binds far more rapidly," yet the association speed was not measured in these experiments, only the dissociation. Extrapolating to mechanisms in living cells (Figure 7) is insufficiently supported by experimental data. The receptor density on cells and their mechanical properties may not validate this model. Without experiments on cells, the model should remain focused on molecular-scale binding.

Reviewer #2

(Remarks to the Author)

In this work, the authors present a structural study on the activity of the binding of hyaluronan (HA) to LYVE-1, a molecular

mechanism critical to the passage of immune cells through the endothelium of lymphatic vessels. In this study, the authors use dynamic force spectroscopy (DFS) in order to propose a “sliding mechanism” by which HA, starting from the end of the chain, will wind through a binding cleft on the surface of the LYVE-1 protein. This mechanism contrasts with the stochastic “sticking” mode observed with the related protein CD44, as well as most observed binding mechanisms between proteins and biopolymer chains. The structure of LYVE-1 is presented for the first time by means of X-ray crystallography, in human and murine apo form as well as the human form bound to an HA octamer and a decamer. The structures reveal a positively-charged cleft to which the HA binds, with multiple water-mediated hydrogen bonds. The authors also characterize the occupancy of these structured waters and examine protein flexibility through molecular dynamics (MD) simulations.

The authors provide very good experimental support for their proposed “sliding” model of binding between LYVE-1 and HA, as well as four high-quality crystal structures. Due to the lack of existing structures and the well-supported and novel nature of the proposed entry mechanism for immune cell entry through the lymphatic endothelium, I believe this work would generate significant interest within the scientific community and marks an important advancement in the understanding of immune system function. Therefore, I recommend publication in Nature Communications following a few minor revisions.

- The authors explain a <10% water occupancy in MD simulations as due to “incomplete sampling” and “inaccuracies in force fields”. This sounds like a cop-out, and additional analysis should be performed to rule out other possibilities – such as significant structural rearrangements occurring between the crystal structure and the simulation. Sometimes crystal structures are contorted, and MD simulations can provide insight if structural re-arrangements were, say, displacing those waters. The structured waters not present in MD might also be due to interactions with neighboring units within the crystal. If insufficient sampling is truly suspected, the authors may consider employing enhanced sampling methods like GaMD to observe the “correct” water occupancy. If poor force fields are truly suspected, then the authors may consider using a more accurate water model, such as TIP4Pew, and re-run the simulations.

- Fig 1C seems to show a slight difference to the shape of the reductionist model in Extended Data Fig2. Can the authors discuss likely causes for this difference?

- In the Molecular Dynamics Simulations Methods section, the authors mention that they cap the termini of LYVE-1, but leave the CD44 N-terminus charged. What was the reason for this? Also the authors should explicitly write out the MD details, instead of refer to a previous study. The protein forcefield version, the glycan forcefield version, the non-bonded cutoff, temperature, ensemble, etc. should all be explicitly stated here.

- Authors should be careful to define all acronyms the first time they are used. For instance, “DC” is never defined, and “MD” isn’t defined until later uses of “molecular dynamics”.

- On page 8, line 203, the authors should say “positive charge” instead of “basic charge”.

Reviewer #3

(Remarks to the Author)

In this project the author uses dynamic force spectroscopy (DFS) analyses to show the sliding motion interaction model between LYVE-1 from human and murine and HA domain compare with the sticking mode between CD44 and HA domain. To indicate the detail of interaction, the author also got very high resolution co-crystallized structure alternative mLYVE-1 construct $\Delta 141$ and the hLYVE-1 construct $\Delta 144$ with a synthetic HA octasaccharide (HA8) and decasaccharide (HA10), respectively. The structure showed us that the buried surface area of LYVE-1 is 14% large than in CD44. Comparison between the LYVE-1 apoprotein and HA bound complexes indicates that docking of the sugar occurs most likely through induced fit, achieved by conformational changes in some key amino acid sidechains within the binding groove. And also because of very high resolution, the author showed us some very important water-mediated hydrogen bonds besides the direct H bond at the protein complex binding interface while this water-mediated hydrogen bonding is not found in CD44.

But I think still this project still had some question:

1, in this project, the author should explain what is the relationship between LYVE-1 of human and murine and why we need the function research and structure detail for both LYVE-1 from human and murine even they are so similar (RMSD = 0.68 Å for 110 C α pairs, 1.04 Å across all 115 pairs)

2, the author solved the structures of both mLYVE1 and hLYVE1 in unliganded states by single wavelength anomalous. Please provide more detail about this.

2, from the Table I, Crystallographic data and refinement statistics, the data looks very interesting

A, in all the dataset for 4 crystal structures, all the unique reflections are such low compare their total reflections are all very high even the multiplicity is not that high, the author could explain this in the method.

B, in all the dataset the CC1/2 are 1, it is a little bit unusual, the author should provide the number of overall and the highest resolution shell.

C, the completeness of 8OX3 is 69.05(6.88) looks very strange. That means the completeness of the high-resolution shell 1.05Å is only 6.88%? how could the author get the such good R and R free, 14.99/16.21 from this kind data? also the Average B-factor, the ligands were double than the protein, if author could provide the omit map of the ligands for HA octasaccharide (HA8) and decasaccharide (HA10) should be good.

Version 1:

Reviewer comments:

Reviewer #1

(Remarks to the Author)

The authors fully addressed my previous comments

Reviewer #2

(Remarks to the Author)

The authors have adequately addressed my concerns, and I believe that this manuscript is acceptable for publication.

Reviewer #3

(Remarks to the Author)

From the respond of the revised manuscript "A novel sliding interaction with hyaluronan underlies the mode of action of LYVE-1, the receptor mediating leucocyte entry and trafficking in the lymphatics" by Professor Jackson and colleagues, The authors had replied all the questions.

there are some mistakes from their crystallographic table in the first version. Actually, that is not big problem with this research, but I am so happy they can find and fix this error, and put more detail how to solve all these structures into the manuscript. As a very high-resolution crystal structure, we need more detail to understand the quality of the structure.

As the relationship between the LYVE-1 of human and murine, the authors still not explained very clear. I suspected the author constructed the protein of human and murine at the same time just in case of can't get the structure from one of them. In fact, the structure of LYVE-1 from human and murine are very similar, if the authors could show a sequence alignment between LYVE-1 of human and murine is better to explain this highly conserved interaction model.

In this study, the author tries to keep 1.05-1.07 Å resolution to show more detail, but from the omit map, we can see the intensity map of the mLYVE-1 HA8 still not that good at the same time the hLYVE1 HA10 is much better, but it is enough to show the shape of ligands.

In this manuscript, the authors show very high-resolution crystal structure for proteins APO form and ligands binding form which is around 1.05 - 1.68 Å resolution to know the important interaction mediated by water molecular. That is why we still need Protein crystallography to support our research.

We thank the Reviewers for their positive and enthusiastic appreciation of our work, and insightful comments and suggestions. We have now responded fully to all of their queries and revised the manuscript accordingly. Changes to the text, including the insertion of four new Figures (8,9,19 and 20), in the revised extended data section (are shown in red in the marked up copies of the revised main manuscript and extended data sections that accompanies this submission, with their page and line numbering indicated in the responses below. For completeness, other minor changes, to correct grammar, typos etc are also marked red. Below, we provide a point-by-point response to the individual comments in which the reviewers words are in black text and ours are in blue.

Reviewer #1 (Remarks to the Author)

In this study, the authors reported the binding of LYVE-1 to HA and presented the first crystal structures of this binding interaction. Based on these structures and AFM data, they proposed a highly unusual sliding mode of ligand interaction. While the study is intriguing, some conclusions seem premature and possibly overstated.

The experimental details are somewhat difficult to follow. The results section should better explain how the LYVE-1 receptors and HA molecules were grafted onto the AFM tip and the surface. Additionally, the experimental methods contain unclear details. For example, the AFM planar substrates section mentions that LYVE-1 can be grafted using a histidine tag, but it is unclear how the NTA surface is prepared. Have QCM-D sensors been used?

Response. We had deliberately refrained from presenting too much technical detail in the main text to provide a narrative that is attractive to a wide readership including non-experts in DFS and surface modification. However, we appreciate that clarity on such detail is of utmost importance in the Methods section and have revised that section according to the Reviewer's suggestions.

LYVE-1 constructs were grafted *via* their polyhistidine tag on two distinct surfaces. In most cases, the surface was a 'His-tag capturing QCM-D sensor' which presented chelating groups for histidine capture on a PEG background. In one instance (Extended Data Figure 5), the surface was a supported lipid bilayer (SLB) presenting $(\text{Ni}^{2+}\text{-NTA})_3$ groups.

We emphasise that the suitability of the his-tag capturing QCM-D sensor for DFS analyses was previously validated for CD44•HA interactions. In Bano *et al.* (2016. *Sci. Rep.* 6:34176), the unbinding mechanics for HA polysaccharides were compared between CD44 immobilised via (i) a polyhistidine tag on his-tag capturing QCM-D sensors, (ii) a polyhistidine tag on SLBs presenting $(\text{Ni}^{2+}\text{-NTA})_3$ groups and (iii), a biotin tag on streptavidin-presenting SLBs. The loading rate dependent mean rupture forces were quantitatively identical in all these cases.

Changes to manuscript.

- In the Methods section (page 31, lines 809-811, we have now clarified the prolonged dwell times used for experiments with HA in loop configuration.
- In the Methods section *Anchorage of receptors to planar substrates* (page 37, lines 920-938), we have now clarified that His-tag capturing QCM-D sensors were mostly used for DFS (alongside QCM-D) analyses, and also in which instances other surfaces were used.
- In each appropriate figure of the manuscript, we have now clarified which particular LYVE-1 construct (see red text) was used in the respective experiments.
- In the Methods section (page 30, lines 739-743, we now provide details on the site-specific biotinylation of LYVE-1 $\Delta 238$ with a biotin/His₁₀ dual tag.
- Within Extended Data Figures 17C and 19 we now provide a complete set of QCM-D data demonstrating stable, functional and specific anchorage of each relevant LYVE-1 construct to each relevant surface.

The noise level observed in the AFM curves is surprisingly low. Theoretically, the noise level is directly related to thermal equilibrium and the spring constant, with the formula $F = (k_b * k_c * T)^{1/2}$. A

quick calculation yields a theoretical noise level (with ideal instruments) of 5 to 15 pN. Some curves presented in the paper have noise levels well below these values (e.g., Figure 1c, first inset; Figure 2B). Could there be an issue with the calibration of the instrument or the cantilever?

Response. We wish to point out that the first inset in Figure 1C does not represent experimental data, but rather the results obtained with the reductionist model. This was already pointed out in the legend of Figure 1C and in the main text. We have now revised the panel to clarify this further.

All experimental DFS data shown (see Figures 1B-C and 2B, E, G; Extended Data Figures 3, 4, 5B-C, 6 and 10B-C), display noise levels in the range of 5 to 15 pN.

Changes to manuscript. We have revised Extended Data Figure 4C to correct the range of scale on the y axis, and clearly illustrate the expected noise level.

The shape of the interaction curve between HA and LYVE-1 is unusual and sometimes seen during the desorption of certain polymers. If the experiments were conducted on QCM-D sensors with poorly characterized chemistry, could the authors repeat the experiments on a custom-made surface with well-defined chemistry, such as a gold-coated surface with NTA alkanethiols?

Response. The experiments had indeed been repeated on a second surface type with well-defined chemistry, as shown in Extended Data Figure 5 and mentioned in the original manuscript (see lines 130f in the revised manuscript).

We here used a supported lipid bilayer (SLB) presenting $(\text{Ni}^{2+}\text{-NTA})_3$ to anchor LYVE-1 *via* its polyhistidine tag, *i.e.*, in the same orientation as on the his-tag capturing QCM-D sensors. Extended Data Figure 5B shows that the shape of the force curve for multivalent LYVE-1•HA interactions is qualitatively similar to Main Figure 1C, indicating that the distinctive nanomechanical property of LYVE-1•HA bonds is independent of the type of surface used. In addition, Extended Data Figure 5C demonstrates that the observed interactions require the presence of both LYVE-1 and HA, and can be blocked with function-blocking antibodies, and thus are specific.

We are confident that the SLB surfaces are chemically well defined. Such surfaces had already been used successfully in previous projects (including to probe HA•receptor unbinding mechanics; Bano *et al.* 2016. *Sci. Rep.* 6:34176). The definition of the surface chemistry here pertains to the defined chemical composition of the SLB (*i.e.*, DOPC and $(\text{Ni}^{2+}\text{-NTA})_3\text{-SOA}$) and the formation of an SLB of good quality (*i.e.*, covering the surface fully with minimal defects). The latter is demonstrated by the small final dissipation shift ($\Delta D < 0.3 \times 10^{-6}$) and a final frequency shift of $\Delta F = -29 \pm 1$ Hz) observed by QCM-D, as shown in Extended Data Figure 18.

Changes to manuscript. We have added two references to the legend of Extended Data Figure 18, to substantiate the interpretation of the QCM-D data in terms of SLB quality.

One puzzling aspect is that at the end of the rupture between HA and the surface, the final event should reflect the rupture of a single bond, similar to what is observed in Figure 2B. This should allow observation of a single rupture between an HA chain and a single receptor.

Response. We agree with the reviewer that a distinctive unbinding event is expected when the HA chain end disengages from a LYVE-1 receptor (as compared to continuous sliding of the HA chain through the receptor's binding groove). Whether this response is detectable in the DFS assay is, however, a different question.

In the following, we will illustrate that the rupture force for unbinding through the chain end may well be too small to be detectable. We estimate the expected force range using the predictions of the Bell-Evans model, $F = \frac{k_b T}{x_\beta} \ln \frac{r x_\beta}{k_{\text{off}} k_b T}$. For unbinding through the chain end, we estimated a zero-force off-rate in the range of $10 \text{ s}^{-1} < k_{\text{off}} \lesssim 10^3 \text{ s}^{-1}$ (see page 6, lines 150-155, and Methods section *Estimation of HA: receptor binding rates from DFS data* (lines 874-878). We do not know the exact bond length in this unbinding scenario, but given the nature of the sliding motion it appears plausible for the bond length to be comparable to the length of the binding groove, and we hence assume $x_\beta =$

4 nm (equivalent to 4 disaccharides). The predicted mean rupture forces as a function of the loading rate for a few relevant zero-force off-rates ($k_{\text{off}} = 10, 100$ and 1000 s^{-1}) are displayed in the figure below. Here, we have considered loading rates up to 10^4 pN/s , as this is the maximal value achieved experimentally for unbinding of HA from HA loops (see Figure 2C), but note that the maximal achievable loading rate in the scenario of end-on binding may be somewhat lower than that. It can be seen that a force range of just a few pN or below is quite plausible for our interaction scenario.

This suggests that the unusual curves may be non-specific interactions. If this is incorrect, the authors need to provide very convincing experiments. For instance, they could create surfaces with varying NTA group densities to control receptor density or use point mutations in the LYVE-1 binding pocket to test specificity.

Response. In our manuscript, we have presented several pieces of data which independently and collectively provide strong evidence for the specificity of the observed interactions. For the convenience of both the editor and the reviewers, we summarise them here:

- i. The observed interactions strictly require the presence of both LYVE-1 and HA, and are completely eliminated when either of the two binding partners is excluded from the assay whilst maintaining all other interaction parameters unchanged; see Extended Data Figure 4C.
- ii. The observed interactions can be blocked with a function-blocking anti-LYVE-1 antibody; see Extended Data Figure 4C.
- iii. The magnitude of the observed interactions monotonically increases with the LYVE-1 surface density, as expected; see Extended Data Figure 4B.
- iv. The observed interactions can be qualitatively reproduced on two surfaces with distinct chemistry: PEG (Figure 1C and Extended Data Figure 4C) vs. SLB (Extended Data Figure 5).
- v. The specificity of the LYVE-1/HA interactions is further supported by our QCM-D control experiments in Extended Data Figures 17 and 18.

Changes to manuscript. We have now amended the main text to emphasise the first two of the above points (page 5/6, lines 127-131); all other points were already explicitly covered.

Regarding the DFS in Figure 2C, only four points were obtained within a very narrow loading rate range. To ensure accurate kinetics extraction, a broader loading rate range should be analyzed (at least over 2-3 log). As it stands, the significance of the Bell-Evans fit slope is questionable. A horizontal fit might also be plausible (meaning non-specific interaction). Could the authors explore a larger range or provide a statistical analysis to evaluate the fit's error and the slope's significance?

Response. Following the reviewer's suggestion, we have acquired new data over a broader range of loading rates. The data cover approximately two orders of magnitude in loading rates (compared to

one order of magnitude for the previous data), which we deem the maximal range feasible with our setup.

The new data show a linear dependence of the mean rupture force on loading rate, consistent with the presence of a single energy barrier, and a fit with the Bell-Evans model provides a zero-force off rate $k_{\text{off}} = (8.7 \pm 9.7) \times 10^{-4} \text{ s}^{-1}$. This value is consistent with the previous data ($k_{\text{off}} = (1 \pm 9) \times 10^{-3} \text{ s}^{-1}$). Reassuringly it also shows a reduced confidence interval, although it should be noted that the magnitudes of the confidence interval and the mean are comparable. We therefore consider that our DFS unbinding data can provide a confident upper limit (rather than mean), leading to $k_{\text{off}} < 0.002 \text{ s}^{-1}$.

To further support our conclusions, we have now additionally analysed the rate of LYVE-1 binding to HA loops in the DFS interaction scenario illustrated in Figure 2A. This analysis is presented in the new Extended Data Figure 8 and led to $k_{\text{on}} = 5.5 \pm 3.0 \text{ M}^{-1}\text{s}^{-1}$. Collectively, the DFS analysis thus provides an affinity estimate $K_{\text{d}} = k_{\text{off}}/k_{\text{on}}$ in the range of a few 100 μM (a full error propagation results in $K_{\text{d}} = 158 \pm 197 \mu\text{M}$, yet given the rather larger error bar 'a few 100 μM ' is a more befitting quantification).

In addition, we have quantified the LYVE-1•HA binding affinity by surface plasmon resonance (SPR) using immobilised LYVE-1 as the ligand and HA decasaccharides as analyte. This analysis is presented in the new Extended Data Figure 9 and gave $K_{\text{d}} = 227 \pm 76 \mu\text{M}$. Whilst the on and off rates were too fast to be measured independently by SPR, the consistency in the K_{d} analysis from DFS and SPR is reassuring and lends confidence to the robustness of the DFS rate analyses.

Changes to manuscript. We have updated Figure 2B-C and Extended Data Figure 7 to provide a dataset covering the largest range of loading rates feasible with our setup. In Figure 2C, we have also revised the presentation of k_{off} to make it clear that the analysis of the DFS unbinding data provides merely an upper boundary to the zero-force off rate. We have also added new Extended Data Figures 8 and 9, and briefly describe these data in the main text (page 6, lines 150-155, and page 36, lines 886-890). Note that we have now added the names of two additional co-authors, Dr. Kalia Cook and Dr. Iain Manfield (University of Manchester) to the authorship list of the revised manuscript, in recognition of their key roles in performing and analysing these new SPR experiments.

We note that the recent DFS data set shows a moderate yet significant increase in the magnitude of the rupture forces compared to the previous data set. The precise reason for this remains unclear, but quite possibly it reflects minor batch to batch variations in the LYVE-1 samples used for the past vs. recent analyses (we did confirm that the cantilever spring constant was calibrated properly in all experiments). To ensure consistency across experiments, we used the same batch of LYVE-1 in the recent DFS and SPR analyses.

The discussion section requires careful revision, as some points are not supported by data. For instance, lines 339-340 state, "our DFS analyses revealed that LYVE-1 binds far more rapidly," yet the association speed was not measured in these experiments, only the dissociation.

Response. In the previous version of our manuscript, we had conservatively estimated the association rate for end-on binding from the DFS data at $k_{\text{on}} > 10^5 \text{ M}^{-1}\text{s}^{-1}$. This remains stated on page 6 line 156 of the revised main text, with details provided in the Methods section *Estimation of HA : receptor binding rates from DFS data* (page 35/36, lines 869-885). In our revised version, we now also provide an estimate of the association rate for side-on binding from the DFS data, at $k_{\text{on}} = 5.5 \pm 3.0 \text{ M}^{-1}\text{s}^{-1}$ (see previous point for details). Jointly, these data fully support our original claim.

Extrapolating to mechanisms in living cells (Figure 7) is insufficiently supported by experimental data. The receptor density on cells and their mechanical properties may not validate this model. Without experiments on cells, the model should remain focused on molecular-scale binding.

Response. We disagree with the reviewer on this point.

Firstly, in order that the findings from our crystallographic and biophysical analyses be appreciated by a wider audience, we feel it is important to provide a clear pictorial image of how they might translate to the physiological situation – the LYVE-1 mediated adhesion and entry of iHA-coated

immune cells to lymphatic vessels. The schematic model we present in Figure 7 provides the first mechanistic explanation for how this could work. We anticipate that the readership of our work will not be confined to biophysicists and structural biologists, but will include many and perhaps even a majority of biologists and immunologists who will have a close interest in the topic. Indeed, in the opinion of Reviewer 2, our results "..... mark an important advancement in the understanding of immune system function".

The schematic model in Figure 7 is based not only on the structural and mechanical properties of HA LYVE-1 interactions at the molecular scale as defined in our present manuscript, but also on our previous observations of such interactions in live cells. These have indicated that the distribution of LYVE-1 on the surface of freshly isolated primary human lymphatic endothelial cells is dynamic, and that its engagement by HA either on the surface of dendritic cells, on latex beads or complexed with known HA-binding partners induces its high density clustering at the HA:LEC adhesive interface (Lawrance, W. et al *J Biol Chem* **291**, 8014-30 (2016); (Johnson, L.A. et al *Life Sci Alliance* **4**, (2021); Stanly, T.A. et al. *J. Biol. Chem.* **295**, 5036-5050 (2020)).

Furthermore, we have imaged this clustering during the interaction of DCs with LECs in real-time by video microscopy, which showed that the adherent DCs actively migrate across the surface of LEC monolayers in a manner that is dependent on LYVE-1:HA binding (Johnson, L.A. et al *Life Sci Alliance* **4**, (2021)) and consistent with the rapid on/off sliding interaction identified in our present manuscript. Although these references are already cited in the Discussion, they may not have been highlighted sufficiently before.

Changes to manuscript. Nevertheless, and in line with the reviewer's concerns, we have changed the title of the Figure from "Schematic model illustrating how the sliding mode of HA adhesion/de-adhesion to LYVE-1 facilitates dendritic cell attachment and entry to lymphatic capillaries" to "**Hypothetical model of how a sliding mode of HA adhesion/de-adhesion to LYVE-1 facilitates dendritic cell attachment and entry to lymphatic capillaries**" in the revised manuscript (page 22, lines 553-554). We hope this will now be acceptable.

Reviewer #2 (Remarks to the Author)

In this work, the authors present a structural study on the activity of the binding of hyaluronan (HA) to LYVE-1, a molecular mechanism critical to the passage of immune cells through the endothelium of lymphatic vessels. In this study, the authors use dynamic force spectroscopy (DFS) in order to propose a "sliding mechanism" by which HA, starting from the end of the chain, will wind through a binding cleft on the surface of the LYVE-1 protein. This mechanism contrasts with the stochastic "sticking" mode observed with the related protein CD44, as well as most observed binding mechanisms between proteins and biopolymer chains. The structure of LYVE-1 is presented for the first time by means of X-ray crystallography, in human and murine apo form as well as the human form bound to an HA octamer and a decamer. The structures reveal a positively-charged cleft to which the HA binds, with multiple water-mediated hydrogen bonds. The authors also characterize the occupancy of these structured waters and examine protein flexibility through molecular dynamics (MD) simulations.

Note to reviewer. We would like to draw attention to the following. To comply with the mandatory Nature Communications "Reliability and reproducibility checklist for molecular dynamics simulations" document, we have carried out two further independent MD runs to corroborate our original computational results and provide statistics. These are now detailed in the "Results" section on pages 12/13, lines 301-312) and in Extended Data Table 4 with its footnotes.

The authors provide very good experimental support for their proposed "sliding" model of binding between LYVE-1 and HA, as well as four high-quality crystal structures. Due to the lack of existing structures and the well-supported and novel nature of the proposed entry mechanism for immune cell entry through the lymphatic endothelium, I believe this work would generate significant interest

within the scientific community and marks an important advancement in the understanding of immune system function. Therefore, I recommend publication in Nature Communications following a few minor revisions.

Response. We thank the reviewer for their enthusiastic appraisal of our manuscript and for their appreciative comments. We are pleased that they consider our work “marks an important advancement in the understanding of immune system function” and excited that it will “generate significant interest within the scientific community”.

- The authors explain a <10% water occupancy in MD simulations as due to “incomplete sampling” and “inaccuracies in force fields”. This sounds like a cop-out, and additional analysis should be performed to rule out other possibilities – such as significant structural rearrangements occurring between the crystal structure and the simulation. Sometimes crystal structures are contorted, and MD simulations can provide insight if structural re-arrangements were, say, displacing those waters.

Response/Changes to manuscript. We take the point expressed by the reviewer that the statement “...occupancies <10% are probably artifacts of MD due to incomplete sampling and inaccuracies in force fields for charged systems ³⁹.” on page 12 (line 296) of our original manuscript may have appeared rather inadequate and simplistic. We have addressed the issue by including an updated, more detailed paragraph on page 13 of the revised manuscript Main text (lines 313-317). As suggested by the reviewer, there are indeed discrepancies between the crystal structures and the MD trajectories. For these highly charged complexes, the reasons are mostly the erroneous overestimation of charged interactions when using non-polarizable force fields (as recently set out in Nencini et al. *J Chem Theory Comput* 20, 7546–59 (2024)). This distorts the crystallographic HA binding mode and prevents the formation of some native H-bonds, both direct and water-mediated. This issue and others is now covered under “Models and parameters” (page 39/40, lines 968-978), along with others, in a much more detailed and expanded description of the methods we used for MD simulations on pages 38-41 in our revised manuscript that includes several new references including Nencini et al, and in the “b” footnote to the Extended Data Table 4.

The structured waters not present in MD might also be due to interactions with neighboring units within the crystal.

Response. We did consider this possibility. However, examination of the crystal packing reveals that none of the water molecules identified as playing a role in the binding of HA to LYVE-1 in the crystal structures is involved in a crystal contact, and hence we have concluded that crystal packing is not a likely explanation for the mismatch between waters observed experimentally and those found in the MD simulations.

Changes to manuscript. We have added the following sentence “Importantly, none of the waters is involved in crystal contacts, precluding the possibility they represent artefacts of crystallisation.” to the paragraph on page 12 (lines 294-295) of the revised Main text to cover this issue.

If insufficient sampling is truly suspected, the authors may consider employing enhanced sampling methods like GaMD to observe the “correct” water occupancy.

Response/Changes to manuscript. We have re-drafted the sentence referring to the “enhanced sampling” (see the response above) because the true reason for the inability of MD to reproduce some (including water-mediated) H-bonds is the erroneous overestimation of charged interactions when using non-polarisable force fields (Antila et al. *J. Chem. Theory Comput.* 2024, 20, 7546–7559, new reference 92) which leads to distortion of the crystallographic HA binding mode.

If poor force fields are truly suspected, then the authors may consider using a more accurate water model, such as TIP4Pew, and re-run the simulations.

Response/Changes to manuscript. The LYVE/HA systems are highly charged and thus very difficult to describe reliably in atomistic detail. We have tried to use the more advanced TIP4P/2005 water model (Abascal and Vega *J Chem Phys* 123, 234505 (2005)) but the crystallographic binding modes were unstable. This is in line with observations in a large-scale MD study of testing multiple water models for protein–glycosaminoglycan complexes (Anila and Samsonov *J Chem Inf Model* 64, 1691 (2024)). We note that a more powerful approach using implicit polarisable force fields has just recently become available. We now mention this approach in a new section entitled “Models and parameters” in our revised manuscript Methods (page 39, lines 968-978) in which we also include the above new references.

- Fig 1C seems to show a slight difference to the shape of the reductionist model in Extended Data Fig2. Can the authors discuss likely causes for this difference?

Response. Whilst the overall curve shapes are broadly similar, the reductionist model predicts a somewhat steeper decrease in the force prior to full disengagement of the LYVE-1 receptors from the HA chain than what is seen in the experiment.

The reason for this is not entirely clear to us, and subject to current investigations. One potential explanation is that a few HA chains (as opposed to a single HA chain) are being pulled simultaneously in the experiments (as mentioned in the legend of Figure 1A and in the Methods section *Anchorage of HA to AFM probes*). Such an effect was not usually observed for the CD44 receptor, but the much higher end-on binding rates of the LYVE-1 receptor make it plausible that more than one HA chain anchored near the AFM tip apex is being engaged by LYVE-1 during the brief presence of the AFM tip near the receptor-covered surface. Because HA chains are anchored at a range of heights (determined by a stochastic binding process) near the AFM tip apex, the effective distance of unbinding (quantified relative to the tip apex location) would vary between chains thus leading to a smearing out of the disengagement response.

We have refrained from discussing this point in the manuscript itself, because it is rather technical and speculative at the moment, and thus risks distracting from the main narrative.

- In the Molecular Dynamics Simulations Methods section, the authors mention that they cap the termini of LYVE-1, but leave the CD44 N-terminus charged. What was the reason for this? Also the authors should explicitly write out the MD details, instead of refer to a previous study. The protein forcefield version, the glycan forcefield version, the non-bonded cutoff, temperature, ensemble, etc. should all be explicitly stated here.

Response/Changes to manuscript. The reasons for capping are now explained in the revised and expanded MD simulations Methods section of the main text within the paragraph entitled “Systems” on page 38/39 (lines 944-955) of the revised manuscript. The additional explicit MD details requested by the reviewer are also included in this new and much expanded part of the Methods section under the titles “Preparation”, “Models and parameters”, “Relaxation”, “Production”, “Analysis” and “Restrained MD” (pages 38-41, lines 956-1009). We apologise for the fact that these were not included in the original version of our manuscript.

- Authors should be careful to define all acronyms the first time they are used. For instance, “DC” is never defined, and “MD” isn’t defined until later uses of “molecular dynamics”.

Response. We apologise for this oversight. Both DC (dendritic cell) and MD (molecular dynamics) are now defined upon their first appearance in the text (page 3, lines 57-58 and page 4, line 87 respectively, of the revised Main text). In addition, to highlight the role of the DCs that migrate via lymphatics in antigen priming and presentation within downstream lymph nodes, we also now describe them as “antigen-loaded dendritic cells” on line 57.

- On page 8, line 203, the authors should say “positive charge” instead of “basic charge”.

Response. We thank the reviewer for pointing out this error. The term “positive charge” has now been substituted in the revised Main text (page 9, line 211 in the revised version).

Reviewer #3 (Remarks to the Author)

In this project the author uses dynamic force spectroscopy (DSF) analyses to show the sliding motion interaction model between LYVE-1 from human and murine and HA domain compare with the sticking mode between CD44 and HA domain.

To indicate the detail of interaction, the author also got very high resolution co-crystallized structure alternative mLYVE-1 construct $\Delta 141$ and the hLYVE-1 construct $\Delta 144$ with a synthetic HA octasaccharide (HA8) and decasaccharide (HA10), respectively. The structure showed us that the buried surface area of LYEV-1 is 14% large than in CD44.

Comparison between the LYVE-1 apoprotein and HA bound complexes indicates that docking of the sugar occurs most likely through induced fit, achieved by conformational changes in some key amino acid sidechains within the binding groove. And also because of very high resolution, the author showed us some very important water-mediated hydrogen bonds besides the direct H bond at the protein complex binding interface while this water-mediated hydrogen bonding is not found in CD44.

But I think still this project still had some question:

1, in this project, the author should explain what is the relationship between LYVE-1 of human and murine and why we need the function research and structure detail for both LYVE-1 from human and murine even they are so similar (RMSD = 0.68 Å for 110 C α pairs, 1.04 Å across all 115 pairs)

Response. Solving both structures and comparing them has revealed key principles which seem to apply to LYVE in distinction to CD44. It is also important to note differences between the murine and human forms, specifically in relation to the gating residues in the HA-binding groove as well as similarities such as the consistent arrangement of water molecules within this groove. The conservation of water molecules between the murine and human forms highlights what the reviewer recognises as critically important contacts between protein and saccharide.

2, the author solved the structures of both mLYVE1 and hLYVE1 in unliganded states by single wavelength anomalous. Please provide more detail about this.

Response. We apologise for the overly brief account of the structural solution. In the revised manuscript we now provide a more detailed description. In fact there was one single wavelength anomalous dataset collected, for human LYVE-1 in an apo/unliganded state. This dataset was collected at a wavelength of 1.77 Å, on beamline i24 at the Diamond Light Source. The structure was solved in phenix.autosol, and following density modification, an initial model was subsequently built. The hLYVE-1 apo structural model was then used in molecular replacement for a higher resolution dataset for the same protein and space group, and also for the mLYVE-1 apo structure. The apo structures of mLYVE-1 and hLYVE-1 were then used to solve the structures of their respective liganded states.

For the Reviewer’s consideration, a snapshot of the experimentally-phased map is shown on the left in the image below, and superimposed with the final refined model on the right.

Changes to manuscript. The new version of the phasing description has been inserted in the Methods section of the revised main text (page 27/28, lines 671-685) as follows: ‘The crystallographic data were indexed and scaled using autoPROC⁶⁵ or Xia2⁶⁶ software. The structure of hLYVE1 was determined in an unliganded state by single wavelength anomalous dispersion, using phenix.autosol⁶⁷, for a dataset that was collected from a single crystal with a spacegroup of P42₁2, at a wavelength of 1.77Å and extending to 1.91Å resolution. Model building was conducted using phenix.autobuild⁶⁸ prior to remodeling and refinement using Coot⁶⁹ and phenix.refine⁷⁰. The structures of the apo forms of hLYVE1 for a new dataset from a second crystal grown in the same conditions (collected at a wavelength of 0.9686Å), and of mLYVE1, were subsequently determined by molecular replacement using Phaser within Phenix^{70,71} and with the experimentally-phased hLYVE1 apo structure as a search model. The mLYVE1 apo structure (residues 29-143 built) was determined at a resolution of 1.54 Å, and the final hLYVE apo structure (residues 24-144) at a resolution of 1.64 Å. Subsequently, structures for mLYVE•HA8 and for hLYVE1•HA10 were determined using molecular replacement with their respective apo state models, again using Phaser within Phenix^{70,71}, prior to model rebuilding and refinement as for the apo states. The data collection and structure refinement statistics for each of the reported structures are summarised in Extended Data Table 1.’

3, from the Table I, Crystallographic data and refinement statistics, the data looks very interesting.

A, in all the dataset for 4 crystal structures, all the unique reflections are such low compare their total reflections are all very high even the multiplicity is not that high, the author could explain this in the method.

Response. We apologise for this error. The wrong total reflection values were given for the Apo datasets in the original version of our manuscript. The correct values are as follows: mLYVE1 Apo 90831; hLYVE1 Apo 303695.

Changes to manuscript. These have now been placed in a revised version of Extended Data Table 1. We think these values now align well with those expected.

B, in all the dataset the CC1/2 are 1, it is a little bit unusual, the author should provide the number of overall and the highest resolution shell.

Response. We thank the referee for pointing this out and for requesting updated statistics. The requested statistics are as follows:

mLYVE1 Apo overall CC1/2 0.999, highest shell 0.954.

hLYVE1 Apo overall CC1/2 1.000, highest shell 0.498.

mLYVE-HA8 overall CC1/2 0.995, highest shall 0.378.

hLYVE-HA10 overall CC1/2 0.999, highest shell 0.683.

Changes to manuscript. These values have now been placed in the revised version of Extended Data Table 1 (overall values shown, with highest shell values in parentheses).

C, the completeness of 8OX3 is 69.05(6.88) looks very strange. That means the completeness of the high-resolution shell 1.05Å is only 6.88%? how could the author get the such good R and R free, 14.99/16.21 from this kind data? also the Average B-factor, the ligands were double than the protein, if author could provide the omit map of the ligands for HA octasaccharide (HA8) and decasaccharide (HA10) should be good.

Response. We thank the referee for highlighting the issue with our data reporting.

The cause of the low completeness of the highest resolution shell is the narrowness of the final shell's resolution range (1.07-1.05 Å) as drawn by Phenix in producing the data statistics. For example, if we reduce the number of shells but maintain the current resolution cutoff of 1.05 Å, then the outer shell covers 1.13-1.05 Å and is 14.68% complete. Alternatively, if we lower the resolution to 1.1 Å then the outer shell is reported as 1.17-1.1 Å and is 28% complete, and if we lower the resolution to 1.2 Å then the equivalent values are 1.28-1.2 Å and 67.81% complete. Overall, the data are 77.75% complete to 1.1 Å and 91.11% complete to 1.2Å. We would prefer to maintain the current reporting but add an explanatory note to Extended Data Table 1.

Changes to manuscript. An explanatory note has now been inserted in the revised version of Extended Data Table I (shown with asterisk).

Additionally, for the consideration of the Reviewer, Omit maps for the human and murine LYVE-1:HA complexes have been calculated and are shown below -

mLYVE-1 HA8 complex composite omit map with refined structure

hLYVE1 HA10 complex composite omit map with refined structure

Point by point response to reviewers

The reviewer's comments are in normal text below, and author's responses are in italics.

All revisions to the text in the manuscript are marked in red type.

Reviewer #1 (Remarks to the Author)

The authors fully addressed my previous comments

Reviewer #2 (Remarks to the Author)

The authors have adequately addressed my concerns, and I believe that this manuscript is acceptable for publication.

Reviewer #3 (Remarks to the Author)

From the respond of the revised manuscript "A novel sliding interaction with hyaluronan underlies the mode of action of LYVE-1, the receptor mediating leucocyte entry and trafficking in the lymphatics" by Professor Jackson and colleagues, The authors had replied all the questions.

there are some mistakes from their crystallographic table in the first version. Actually, that is not big problem with this research, but I am so happy they can find and fix this error, and put more detail how to solve all these structures into the manuscript. As a very high-resolution crystal structure, we need more detail to understand the quality of the structure.

We are grateful to the Reviewer for acknowledging our corrections of mistakes in the originally submitted version of the crystallographic table. We too are very happy to have had the chance to do this and are glad the reviewer raised the points they did. We agree that, as a very high-resolution structure, our results needed full and accurate documentation.

As the relationship between the LYVE-1 of human and murine, the authors still not explained very clear. I suspected the author constructed the protein of human and murine at the same time just in case of can't get the structure from one of them. In fact, the structure of LYVE-1 from human and murine are very similar, if the authors could show a sequence alignment between LYVE-1 of human and murine is better to explain this highly conserved interaction model.

We accept that the relationship between murine and human versions of LYVE-1 could have been better explained. To address this perceived shortcoming we have amended the text on page 8, lines 184-188, as follows "As highlighted by the sequence alignments in Supplementary Figure 1, there is a high degree of homology between mLYVE-1 and hLYVE-1 as well as a significant difference in the identities of some ligand binding residues. To enable us to investigate these features more fully and obtain a comprehensive picture of the LYVE-1•HA binding interaction we carried both LYVE-1 species forward for structural analyses." Regarding the request to include a sequence alignment of human and murine LYVE-1, we already had provided just such an alignment both in the original and recently revised

versions of our manuscript in the form of Supplementary Figure 1. However, to avoid this being overlooked by other readers, we have now highlighted the presence of the sequence alignment by adding two further citations to Supplementary Figure 1 on lines 196 and 200 of page 8 in the final revised text.

In this study, the author tries to keep 1.05-1.07 Å resolution to show more detail, but from the omit map, we can see the intensity map of the mLYVE-1 HA8 still not that good at the same time the hLYVE1 HA10 is much better, but it is enough to show the shape of ligands.

We have acknowledged this paucity of data in the highest resolution shell for mLYVE1-HA8 in the note added to the crystallographic table and have now complemented this with inclusion of the omit maps for the HA ligands for mLYVE1 and hLYVE1 in a new Supplementary Figure 12.

In this manuscript, the authors show very high-resolution crystal structure for proteins APO form and ligands binding form which is around 1.05 - 1.68 Å resolution to know the important interaction mediated by water molecular. That is why we still need Protein crystallography to support our research.

We are grateful to the reviewer for their closing, highly affirmatory comment on the importance of work such as this, and of protein crystallography in general.

In conclusion, we thank all three reviewers for their helpful and supportive comments, which have greatly enhanced our manuscript.